# Mesenchymal stromal cell activation by breast cancer secretomes in bioengineered 3D microenvironments

Ulrich Blache[1,2,*] , Edward R Horton[3,*] , Tian Xia[3], Erwin M Schoof[4], Lene H Blicher[4], Angelina Schönenberger[2,5], Jess G Snedeker[2,5], Ivan Martin[6], Janine T Erler[3], Martin Ehrbar[1]

Mesenchymal stromal cells (MSCs) are key contributors of the tumour microenvironment and are known to promote cancer progression through reciprocal communication with cancer cells, but how they become activated is not fully understood. Here, we investigate how breast cancer cells from different stages of the metastatic cascade convert MSCs into tumour-associated MSCs (TA-MSCs) using unbiased, global approaches. Using mass spectrometry, we compared the secretomes of MCF-7 cells, invasive MDA-MB-231 cells, and sublines isolated from bone, lung, and brain metastases and identified ECM and exosome components associated with invasion and organ-specific metastasis. Next, we used synthetic hydrogels to investigate how these different secretomes activate MSCs in bioengineered 3D microenvironments. Using kinase activity profiling and RNA sequencing, we found that only MDA-MB-231 breast cancer secretomes convert MSCs into TA-MSCs, resulting in an immunomodulatory phenotype that was particularly prominent in response to bone-tropic cancer cells. We have investigated paracrine signalling from breast cancer cells to TA-MSCs in 3D, which may highlight new potential targets for anticancer therapy approaches aimed at targeting tumour stroma.

## Introduction

Breast cancer is the most common type of cancer in women, accounting for 30% of cancer cases globally. In particular, breast cancer cells often invade surrounding primary site stroma, enter the vasculature or lymphatic system, and metastasise to secondary organs, leading to worse clinical outcomes for patients (1). Cancer progression is a complex multistep process that is dependent on both the behaviour of cancer cells themselves and the function of nonmalignant support cells in the tumour microenvironment (TME)

(2). Tumour-associated mesenchymal stromal cells (TA-MSCs) are a major component of the TME and are a potential source of cancer-associated fibroblasts (CAFs) (3, 4, 5, 6). TA-MSCs assist cancer progression by promoting metastasis, tumour vascularisation, and immunosuppressive conditions (7, 8, 9, 10, 11). TA-MSCs have been show to promote breast cancer cell malignancy (12, 13, 14) and contribute to cancer cell resistance to chemotherapy (15). Therefore, TA-MSCs and their derived factors are considered as emerging targets for novel anticancer therapies (16). In this regard, several agents targeting tumour stroma are in clinical trials (reviewed in reference (15)).

The TME is a 3D entity made up of multiple cell types and the ECM. It has been shown that cell signalling and drug responses differ when cells are cultured on rigid 2D substrates or using 3D cell culture systems that more closely mimic the TME (17, 18, 19, 20). The vast majority of 3D cancer models are based on animal or tumour-derived ECM components such as collagen, fibrin, and Matrigel hydrogels. In contrast, synthetic hydrogels are a useful alternative when focusing on cell signalling events as they are generated with defined biochemical and biophysical properties and are free of confounding ECM or signalling proteins that are present in ECM-derived hydrogels (21, 22, 23). We have previously developed cytocompatible enzymatically cross-linked poly ethylene glycol (PEG) hydrogels that are matrix metalloproteinase (MMP)-degradable and contain the cell adhesion site arginylglycylaspartic acid (RGD), thereby providing ECM-mimicking microenvironments (24, 25). These biomimetic PEG hydrogels are highly suitable for 3D culture of MSCs and can closely mimic their niches (26, 27).

It has been described that MSC conversion into TA-MSCs occurs via paracrine signalling with breast cancer cells (4, 6); however, the regulation of this unfavourable conversion remains incompletely understood because of the complexity of the underlying molecular events. Here, we apply soft PEG hydrogels (470 Pa) to investigate TA-MSC activation induced by breast cancer cells in an unbiased and comprehensive manner in 3D microenvironments. We analyse breast cancer cell secretomes by mass spectrometry to identify

[1]Department of Obstetrics, University and University Hospital of Zurich, Zurich, Switzerland   [2]Institute for Biomechanics, Eidgenössische Technische Hochschule Zurich, Zurich, Switzerland   [3]Biotech Research and Innovation Centre, University of Copenhagen, Copenhagen, Denmark   [4]Department of Biotechnology and Biomedicine, Technical University of Denmark, Lyngby, Denmark   [5]Biomechanics Laboratory, Balgrist University Hospital, University of Zurich, Zurich, Switzerland   [6]Department of Biomedicine, University Hospital Basel, University of Basel, Basel, Switzerland

Correspondence: janine.erler@bric.ku.dk; martin.ehrbar@usz.ch
*Ulrich Blache and Edward R Horton contributed equally to this work

factors encountered by MSCs; we profile kinase signalling in MSCs to determine how MSCs respond to breast cancer cell-derived factors; and we investigate the reprogramming of MSCs into TA-MSCs by whole-genome RNA sequencing. We performed these analyses using human breast cancer cells with a range of metastatic abilities and found that only MDA-MB-231 secretomes led to the conversion of MSCs into TA-MSCs, which may give insights into the role of TA-MSCs in cancer progression.

# Results and Discussion

## Comparative mass spectrometry analysis of breast cancer secretomes

To investigate MSC activation by cancer cell secretomes from different stages of human breast cancer progression, we used conditioned medium (CM) from a panel of human breast cancer cells (Fig 1A). The pair of MCF-7 and MDA-MB-231 cells are widely used to study breast cancer metastasis as MCF-7 represent breast cancer cells with low invasion capacity, whereas MDA-MB-231 (MDA-WT) are highly invasive. In addition, to investigate whether MSC activation occurs in an organ-specific manner during metastasis, we used MDA-MB-231 sublines derived from organ-specific metastases (MDA-Bone, MDA-Lung, and MDA-Brain). These MDA-MB-231 sublines have been generated by repeated, organ-tropic in vivo selection of MDA-MB-231 cells that had been injected to the left ventricle of nude mice (28, 29, 30, 31, 32). CM was collected from cancer cells after 24 h of serum-free growth and analysed for protein composition by mass spectrometry. Overall, we found 292 proteins that were significantly different between secretomes from MDA-WT and MCF-7 cells, of which 164 proteins were higher in MDA-WT and 128 were higher in MCF-7 (Fig S1A and Table S1). When analysed for overrepresented Gene Ontology (GO) terms, the most significantly overrepresented GO term in both secretomes is extracellular exosome (Fig 1B and C). On further analysis, we identified 173 altered exosome components in total, of which 97 and 76 were enriched in MDA-WT and MCF-7 secretomes, respectively (Fig S1B). Both secretomes clearly differ from each other by the fact that cell–cell adhesion terms were present only in top MCF-7 GO terms (Fig 1B), whereas ECM terms were present only in top MDA-WT GO terms (Fig 1C). Therefore, we next extracted ECM and secreted proteins reported in the human matrisome database (33) and found that 23 ECM proteins were higher in the MDA-WT secretome, whereas only five were higher in the MCF-7 secretome (Fig 1D), which could be attributed to the different EMT statuses that MDA-WT and MCF-7 have. To further subclassify the secretome composition, we divided these proteins into core matrisome and matrisome associated. We found several typical mesenchymal-like ECM proteins, including collagens (COL6A1 and COL12A1) and glycoproteins (FN1, TNC, and laminins [in particular laminin 511]) enriched in the MDA-WT secretome. Interestingly, the two most increased proteins in the MDA-WT secretome (SERPINE1 and MMP1) are both ECM regulatory enzymes known to be involved in breast cancer and metastasis (34, 35). We also identified proteins uniquely in either the MDA-WT (100 exosome proteins, 56 matrisome

proteins) or MCF-7 secretome (43 exosome proteins and 7 matrisome proteins) (Fig S2A and B).

Next, we compared the secretomes of MDA sublines against the parental MDA-WT secretome (Fig 1E and Table S1). We found 68 and 64 exosome or matrisome proteins different for MDA-Bone and MDA-Lung secretomes, respectively. Moreover, although organ-specific secretomes generally produced distinct secretomes, when comparing the matrisome proteins of MDA-Bone and MDA-Lung to MDA-WT we saw similar changes, as seven and two matrisome proteins were commonly up- and down-regulated, respectively (up-regulated: PLOD3, PLOD1, CTSD, LAMB1, LAMA5, TGFBI, and SEMA4B; down-regulated: MMP1 and IGFBP3). We observed that MMP1, which was the most strongly increased protein in MDA-WT versus MCF-7 analysis (Fig 1D), was by far the most decreased protein for both MDA-Bone and MDA-Lung secretomes (Fig 1E). In contrast to MDA-Bone and MDA-Lung, only two proteins were altered in the MDA-Brain secretome (COL6A1 and GGCT). These data demonstrate that MDA-WT cells that have metastasised to bone and lung but not to brain alter their secretome profiles in an organ-specific manner, which might be a consequence of cell adaptation to their new microenvironment in the secondary organ. The lack of change of the MDA-Brain secretome compared with the MDA-WT might be linked to these cells crossing the blood–brain barrier, which then blocks factors in the circulation (36), such as secreted molecules and exosomes released by the primary tumour, reaching these cells or those in the local microenvironment so that MDA-Brain cells are consequently not altered. We also identified several uniquely expressed proteins in the MDA-organ secretomes (Fig S2C–E).

To validate our proteomics data, we analysed the secretomes for three core matrisome proteins (THBS1, FN1, and COL6A1) and three matrisome-associated proteins (MMP1, SERPINE1, and TIMP2) by Western blot analysis (Fig 1F). We were able to confirm the mass spectrometry results between MDA-WT and MCF-7 secretomes for all tested proteins. Similarly, the Western blot analysis also confirmed some of the differences between MDA sublines for these proteins.

## MSC activation by cancer secretomes in bioengineered 3D microenvironments

To resemble the characteristics of the TME under physiologically relevant and highly controlled in vitro conditions, we engineered fully defined 3D microenvironments using PEG hydrogels (Fig 2A). PEG hydrogels were generated by enzymatic cross-linkage of two different star-shaped 8-arm PEG precursor molecules that are end-functionalized by substrate sequences for the transglutaminase FXIIIa (24, 25). To render this inert hydrogel material bioactive, an MMP-sensitive degradation domain in one of the PEG precursor molecules allows matrix remodelling and the fibronectin-derived integrin adhesion ligand RGD enables cell adhesion. This biomimetic microenvironment is highly suitable for the 3D culture of MSCs, which are 3D encapsulated into hydrogels during polymerization. MSCs interact with the microenvironment, spread in 3D, and further remodel it by depositing their own ECM (Fig 2A).

To systematically address the activation of MSCs, we embedded MSCs from four human donors into PEG hydrogels, cultured them

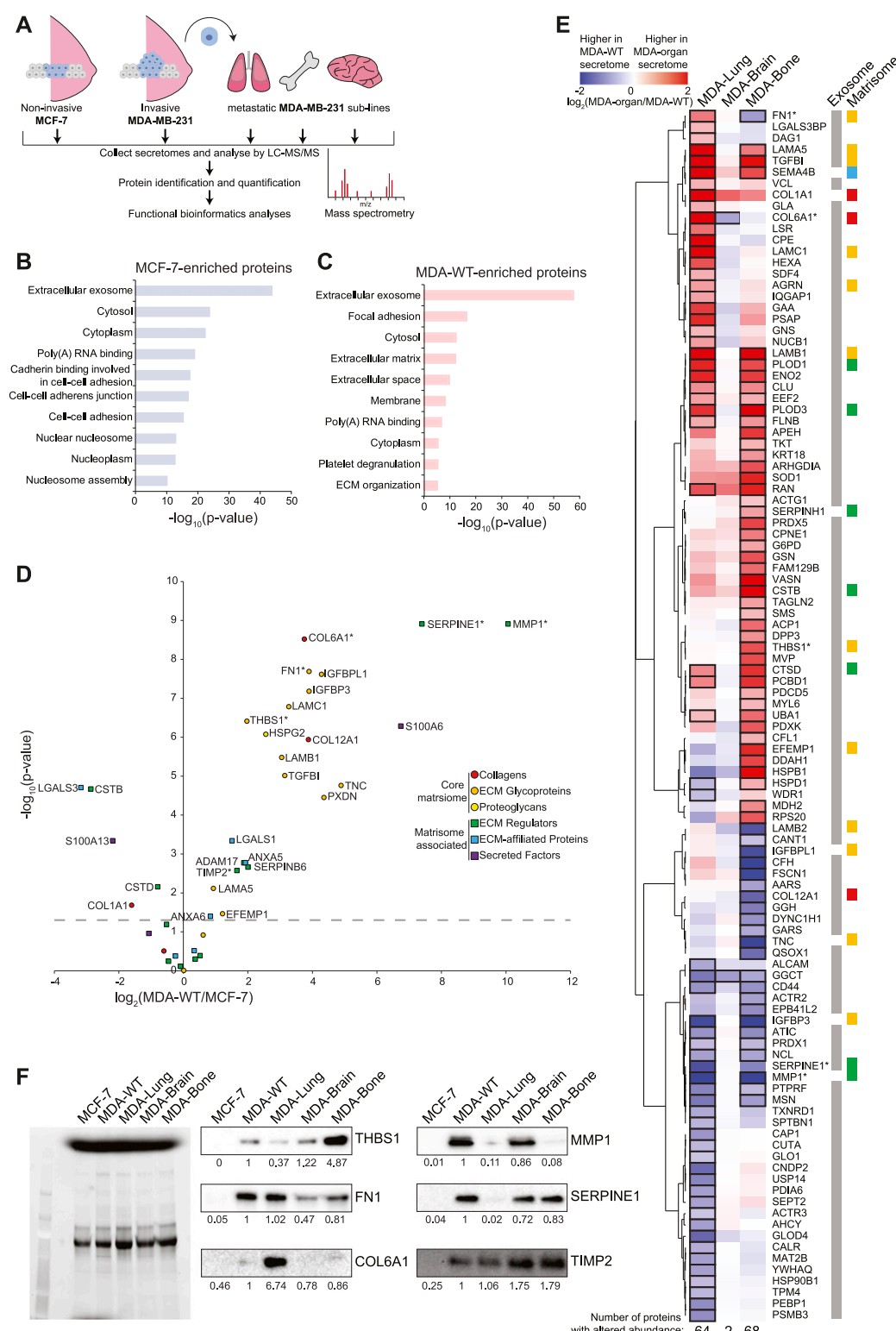

**Figure 1. Comparative proteomic analysis of breast cancer secretomes by mass spectrometry.**
**(A)** To cover a range of cancer progression steps, we used secretomes from noninvasive MCF-7 cells, invasive MDA-MB-231 cells (MDA-WT), and MDA-MB-231 sublines that were isolated from metastatic organs (the lung, bone, and brain). **(B, C)** The 10 most significantly overrepresented gene ontology terms for proteins that were significantly enriched (corrected $P$-value < 0.05) in secretomes of MCF-7 (B) or MDA-WT (C) are listed. **(D)** Subplot comparing matrisome proteins in the secretomes of MDA-WT and MCF-7 cells. **(E)** Hierarchical clustering analysis of significantly changed secretome proteins between MDA-WT and organ-tropic cells. Significantly changed proteins for each comparison are indicated by a black border. Quantitative heat map displays the level of enrichment between the MDA-WT (blue) and organ-tropic (red) secretomes. **(F)** Total protein gel (left) and Western blot analysis (right) of breast cancer secretomes. Band pixel intensities relative to MDA-WT are displayed. Proteins

for 7 d in the presence of breast cancer secretomes, and analysed MSC responses by two omics methods (Fig 2B). First, we investigated downstream signalling events by analysing the kinome of retrieved MSCs using kinase profiling microarrays. These microarrays contain phosphopeptides and based on their phosphorylation levels, the activity of upstream kinases is inferred (37). The relative intensities of tyrosine phosphopeptides and the corresponding upstream phosphotyrosine kinase (PTK) activity values between MDA-WT and MCF-7 are displayed (Fig 2C and D and Tables S2 and S3). Overall, we found that the MDA-WT secretome strongly activates multiple PTKs in MSCs compared with the MCF-7 secretome. The most significantly and intensely changed PTKs are type A ephrin receptors (EPHA1, EPHA3, EPHA4, and EPHA8), which are known to be involved in cancer and angiogenesis (38, 39). Moreover, other strongly activated PTKs are also related to angiogenesis (VEGFR2 and FLT4 [VEGFR3]) or are prominent proto-oncogenes (KIT, FYN, SRC, MET, and ABL1). We observed fewer changes in serine/threonine phosphorylation levels (Fig S3A), which resulted in eight activated and 10 deactivated serine/threonine kinases (STKs) by the MDA-WT secretome (Fig S3B and Tables S4 and S5). When we next investigated altered MSC kinase activity by secretomes of MDA sublines, we found that the MDA-Bone secretome led to increased phosphorylation of VEGFR3 (Fig S4A), and we observed an additional activation of seven PTKs and a reduction in the activity of only two PTKs (Fig 2E and Tables S2 and S3). In contrast, the secretomes of MDA-Lung and MDA-Brain reduced tyrosine phosphorylation levels and thus deactivated many PTKs in MSCs, many of which were among the top activated PTKs by the MDA-WT secretome compared with MCF-7 (type A ephrin receptors, VEGFR2, KIT, FYN, and SRC). The MDA-Bone and MDA-Lung secretomes led to inhibition of several cell cycle–related STKs, including cyclin-dependent (CDK1, CDK5, and CDK9) and checkpoint (CHEK1 and CHEK2) kinases, whereas members of the MAP kinase pathway were differentially activated by different organ MDAs (Figs S3C and S4B and Tables S4 and S5). These data show that kinase signalling pathways are altered in MSCs in response to breast cancer secretomes.

## Secretomes from invasive breast cancer cells induce reprogramming of MSCs into TA-MSCs

To investigate the molecular reprogramming of MSCs by the different breast cancer secretomes, we performed an unbiased transcriptome analysis using RNA sequencing followed by differential gene expression analysis. RNA sequencing is beneficial over protein analysis as we are able to ensure that any observed changes are occurring in the MSCs and are not simply changes in the breast cancer secretomes themselves. In addition, it allows quantification of transcripts that might not encode for functional proteins, which would be missed by proteome analysis. Surprisingly, no gene transcript was significantly altered in MSCs by the MCF-7 secretomes compared with serum-free control medium (Table S6). Consequently, we saw largely the same outcome in transcriptome response if we compared MDA-WT with serum-free

controls or MDA-WT with the MCF-7 secretome. When comparing the MDA-WT and MCF-7 data, 310 gene transcripts were differentially expressed in MSCs (71 MCF-7 enriched, 239 MDA-WT enriched; Table S6). The top 10 most differentially expressed transcripts were all up-regulated by the MDA-WT secretome and consisted of MMPs (MMP3 and MMP13), secreted proteins (complement C3 and the chemokine CXCL1), transporters (SLC39A14 and SLC16A3), PGK1, NFKBIZ, GCH1, and CHI3L2. To reveal pathways regulated by breast cancer secretomes in MSCs, we functionally enriched genes using the GO domain Biological Process and we found 362 Biological Process terms containing proteins that were altered in MSCs by the MDA-WT secretome (see Table S7). By hierarchical clustering (Fig 3), we found that prominent clusters show immune signal reception and processing by MSCs (interleukin signalling, immune response signalling, immune activation, and leukocyte activation), in addition to clusters showing the release of immunomodulatory signals by MSCs (cytokine signalling, cytokine production, JAK-STAT, and macrophage/leucocyte differentiation). As well as an immunomodulatory phenotype, we found MDA-WT–induced clusters that comprise transcription, anti-apoptosis, kinase signalling, hypoxia, glycolysis, ion transport, cell migration, and angiogenesis.

Only four GO terms were found when data were analysed within the domain Cellular Component (Fig 4A), which relate to the ECM, the perinuclear cytoplasm, and the NF–kB complex. We identified 10 overrepresented molecular function terms (Fig 4B), such as integrin binding and ion transmembrane transport (solute carrier family SLC). Five terms formed a very prominent cluster referring to cytokine/chemokine activity that included the chemokines CXCL1-3, CXCL5, CXCL6, CXCL8, CCL2, and CCL7; some of which have been previously reported for TA-MSCs (40, 41) and underlines their immunomodulatory reprogramming. Furthermore, the metal-loendopeptidase activity term (MMP1, MMP3, MMP8, MMP9, and MMP13) indicates a matrix-remodelling phenotype of MSCs, which expands earlier findings on the induced proteolytical and thereby promigratory activity of TA-MSCs/CAFs (41, 42). It is also worth noting that the most strongly induced genes in MSCs by the MDA-WT secretome, mainly chemokines and MMPs, are MSC secretion molecules. We have confirmed the increased expression of chemokines and MMPs in MDA-WT–educated MSCs by quantitative real-time PCR (qRT–PCR) (Fig S5A). Further studies are required to elucidate how breast cancer cell–induced MSC secretion factors alter reciprocal signalling in breast cancer cells and the TME. In this regard, it was recently shown that biomechanical alterations in the MSC microenvironment led to increased expression of prosaposin, which increases breast cancer proliferation (43).

Next, we compared the MDA subline and MDA-WT secretomes by investigating their effect on MSCs transcriptome changes (Table S6). Overall, all three MDA subline secretomes were very similar in inducing immunomodulatory MSC reprogramming that was induced by the MDA-WT secretome. When we compared MDA subline secretomes with MDA-WT, we found that the MDA-Bone secretome resulted in additional expression changes in 14 genes (Fig 4C). These include the genes MMP3, SLC39A14, and PGK1 that were

tested in Western blot analysis are highlight by an asterisk in (D, E). In (D, E), data are from three independent experiments and values represent the mean log$_2$ fold change. Protein lysates used in (F) were pooled from three independent experiments.
Source data are available for this figure.

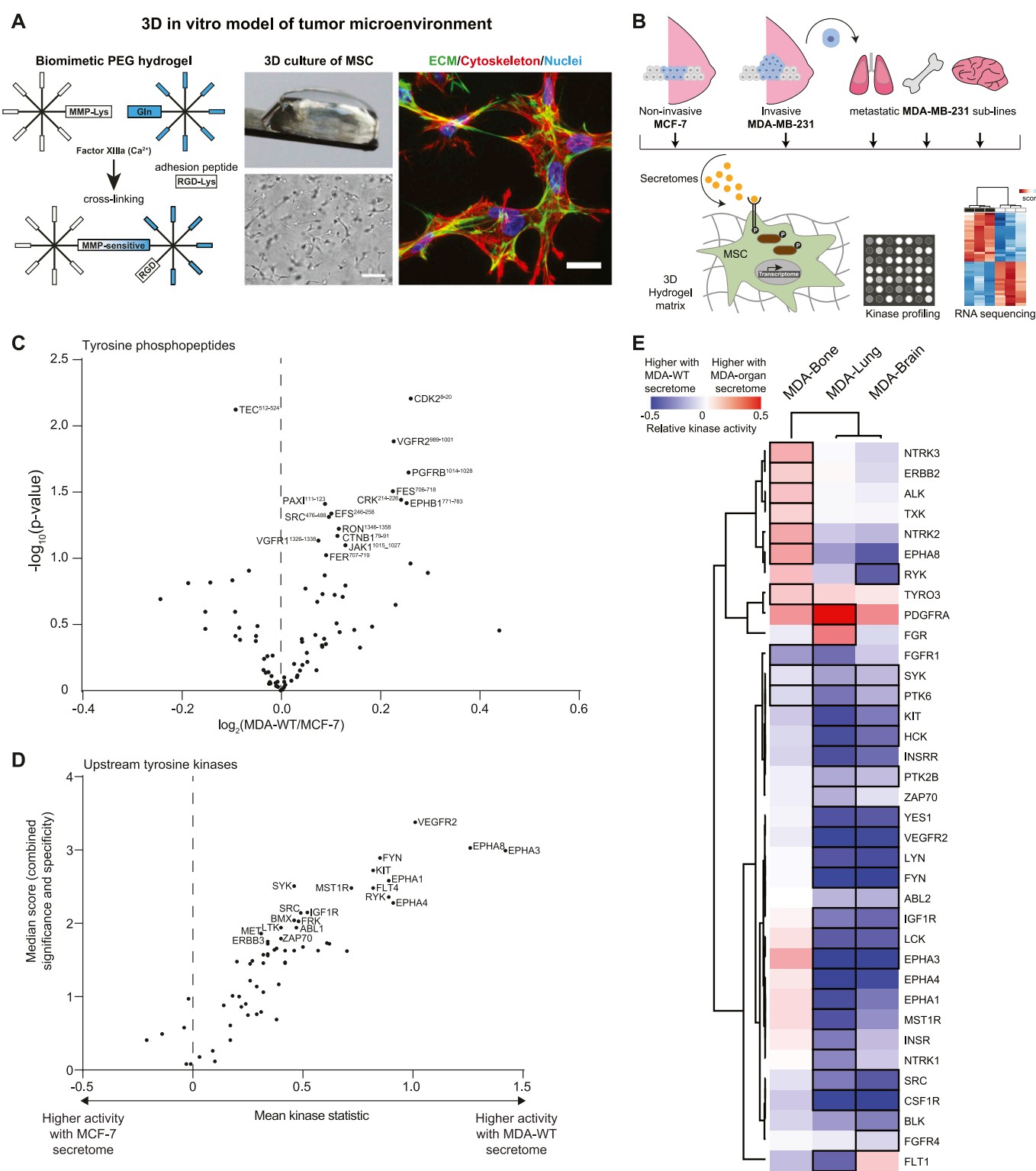

**Figure 2. Bioengineered hydrogels and kinome profiles of MSC responses to breast cancer secretomes.**
**(A)** Schematic of biomimetic PEG hydrogels. Microscopy images show MSCs after 7 d of culture. Bight field image; scale bar: 100 μm. Representative immunofluorescence image of MSCs (F-actin cytoskeleton, red; nuclei, blue) and cell-derived ECM components (Perlecan, green). Image depicts Z-projections through 54 μm. Scale bar 20 μm. **(B)** MSCs were cultured in 3D hydrogels in the presence of secretomes from MCF-7, MDA-WT, or organ-tropic cells. MSCs were analysed by kinase profiling and RNA sequencing. **(C)** Volcano plot comparing phosphorylation levels of tyrosine peptides in MSCs treated with MDA-WT or MCF-7 secretomes. Values represent $\log_2$ intensity values, and the most significantly changed peptides are labeled by gene name and start-end residues of the peptide. **(D)** Comparative analysis of tyrosine kinase activity in MSCs treated with MDA-WT or MCF-7 secretomes. A relative measure of kinase activity (kinase statistic) is plotted against a combined score of significance and

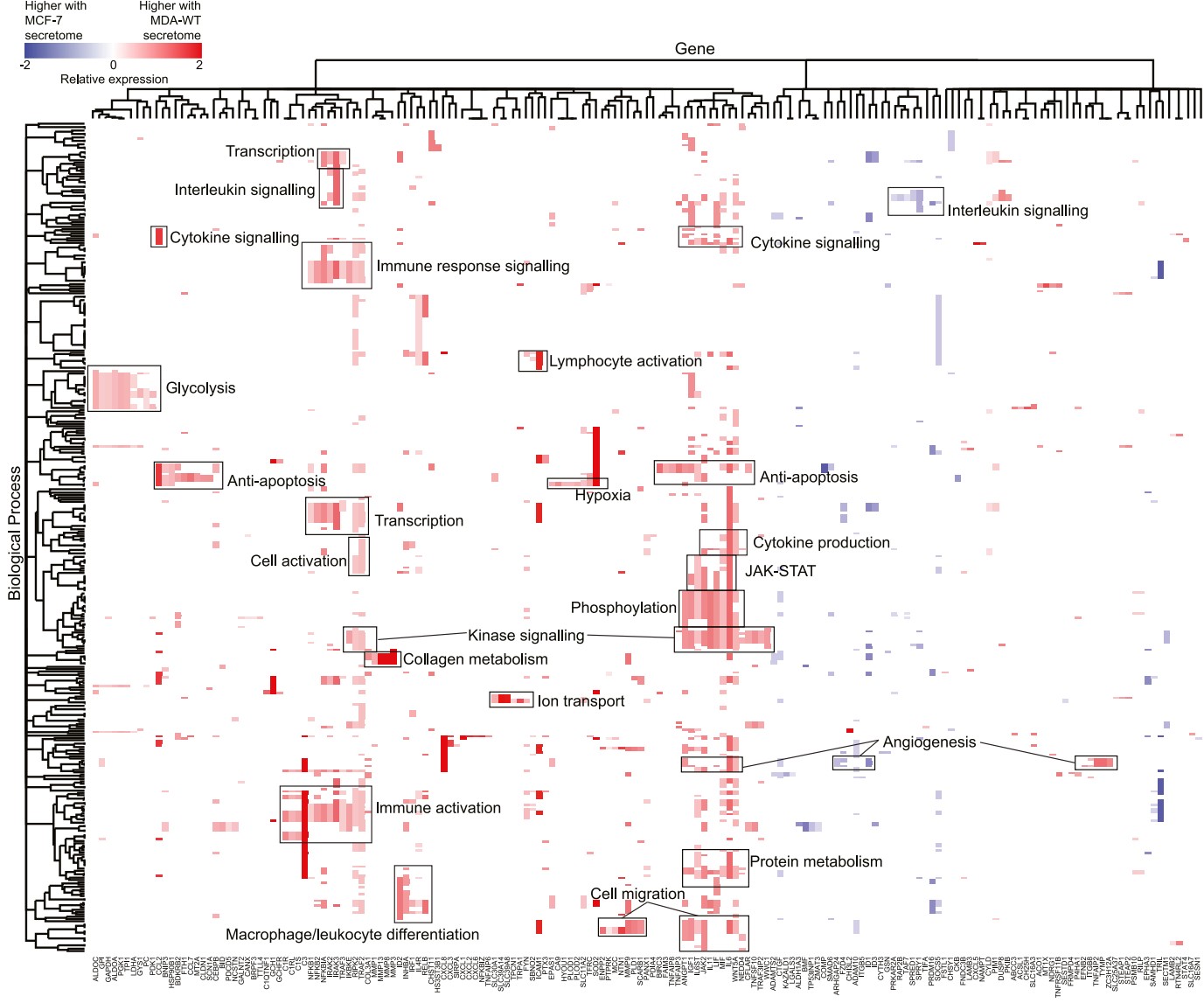

**Figure 3. Transcriptome profiles of MSC responses to breast cancer secretomes.**
Overrepresented Biological Process GO terms from genes that were significantly differentially expressed (*P*-value < 0.05) in MSCs upon treatment with the MDA-WT secretome compared with the MCF-7 secretome are displayed. Terms were hierarchically clustered, which identified clusters of genes associated with a similar set of functional terms. Selected clusters are labeled and a full list of terms is provided in Table S7. Relative expression is displayed as $\log_2(\text{MDA-WT}/\text{MCF-7})$ and colour is proportional to the level of gene expression change. Data represent the mean values from four independent experiments using different donor MSCs (see Table S6).

among the top 10 genes that increased in response to the MDA-WT versus MCF-7 secretomes. In view of their shared functions, it is noticeable that the zinc-dependent MMP3 and the zinc transporters SLC39A14 and SLC39A8 are up-regulated together. Furthermore, five gene transcripts additionally induced by the MDA-Bone secretome all encode for components that are linked to the anaerobic glycolysis pathway (SLC16A3 [MCT4], PGK1, LDHA, PFKP, and TPI). A high aerobic glycolytic activity is a hallmark of cancer and is known as

the Warburg effect in cancer cells (44, 45). Here, we have found an induction of glycolysis, lactate production (LDHA), and lactate transport components, such as SLC16A3, in MSCs by paracrine signalling from invasive cancer cells. This so-called reversed Warburg effect has been described for CAFs and has been found to be a consequence of bidirectional signalling between cancer and stromal cells in the TME (46). Interestingly, we found that this effect was particularly prominent in MSCs in response to the MDA-Bone

specificity for each kinase. **(E)** Hierarchical clustering analysis of tyrosine kinases in MSCs with altered activity in response to MDA-WT and organ-tropic secretomes. Kinases with highest score (>1), indicating altered kinase activity, are displayed with a black border. Quantitative heat map displays relative kinase activity (see Table S3). In all comparisons, data are from at least three independent experiments using different donor MSCs.

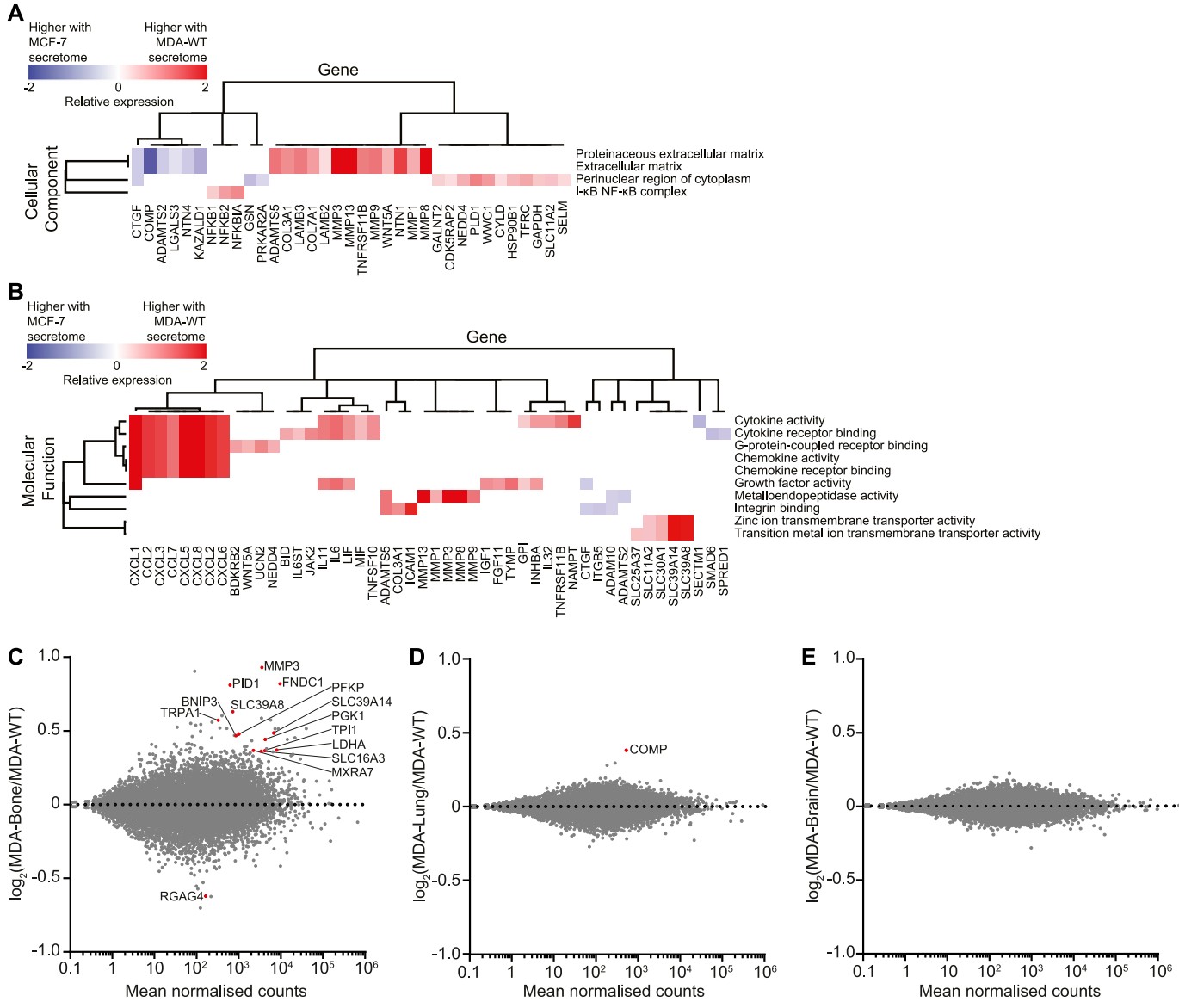

**Figure 4. Additional analysis of transcriptome profiles of MSC responses to breast cancer secretomes.**
**(A, B)** Overrepresented Cellular Component (A) and Molecular Function (B) GO terms from genes that were significantly differentially expressed (*P*-value < 0.05) in MSCs upon treatment with the MDA-WT secretome compared with the MCF-7 secretome are displayed. Terms were hierarchically clustered, which identified clusters of genes associated with a similar set of functional terms. Relative expression is displayed as log₂(MDA-WT/MCF-7) and colour is proportional to the level of gene expression change. **(C–E)** MA plots comparing transcriptome responses in MSCs treated with MDA-WT or organ-tropic secretomes (C, MDA-Bone; D, MDA-Lung; E, MDA-Brain). Significantly changed transcripts are indicated by red circles and are labeled with gene names. In all comparisons, data represent the mean values from four independent experiments using different donor MSCs (see Table S6).

secretome, indicating that this metabolic switch may play a role in bone metastasis. We have validated differences between MSCs treated with MDA-WT and MDA-Bone secretomes for several target genes by qRT–PCR (Fig S5B).

In addition to MDA-Bone, the MDA-Lung secretome led to an increase in the expression of COMP (Fig 4D). In support of a possible function of COMP in metastasis, COMP has been shown to promote lung metastasis of breast cancer and has been found to be the most up-regulated protein in stromal cells at sites of metastases versus primary tumours in ovarian cancer (47, 48). No changes were observed for the MDA-Brain secretome (Fig 4E).

## TA-MSCs do not have altered migration capacity but activate macrophages

Using unbiased, molecular screening approaches, we have shown that MDA-MB-231 secretomes generate a TA-MSC phenotype and have provided a resource of molecular data to the scientific community. To show the further utility of our multiomics data to others, we sought to demonstrate several ways that the datasets can be used to generate hypotheses regarding the role of MSCs in cancer, focusing on the MDA-WT and MCF-7 secretomes as this is where we observed the largest molecular differences in MSCs.

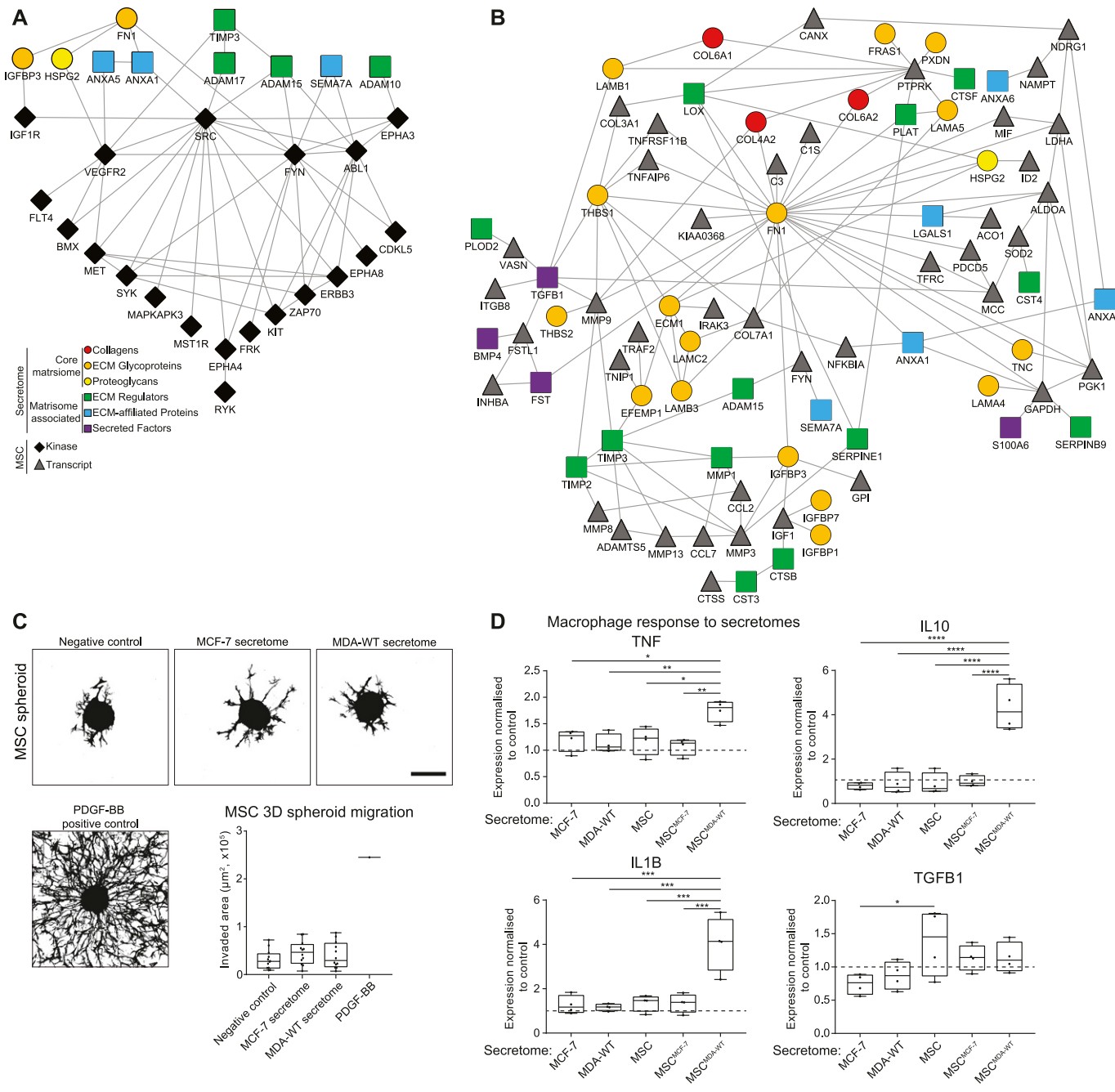

**Figure 5. Network analysis, 3D migration, and immunomodulatory capacity of TA-MSCs.**
**(A, B)** Networks show matrisome proteins enriched in the MDA-WT secretome versus MCF-7 and kinases activated (A) or transcripts up-regulated (B) in MSCs by the MDA-WT secretome versus MCF-7. In both networks, node shape and colour corresponds to protein category as indicated and unconnected proteins are not shown. **(C)** Spheroid-based MSC migration experiments in PEG hydrogels. Z-stack projections of F-actin–stained MSC spheroids treated with breast cancer secretomes or PDGF-BB. Scale bar: 200 μm. Quantification of invaded area: Box plot shows the median (line), 25th and 75th percentiles (box), and min/max (whiskers). N = 12 spheroids from three independent experiments using three different donors. Individual data point for PDGF-BB. **(D)** Macrophage activation assay. Gene expression of macrophages treated with breast cancer secretomes (MCF-7, MDA-WT) or with secretomes collected from MSCs that were pretreated with breast cancer secretomes (MSC, MSC^MCF-7, and MSC^MDA-WT). Gene expression was analysed by qRT–PCR and normalized on two reference genes (GAPDH and ACTB). Dashed line: untreated control. Box plot shows the median (line), 25th and 75th percentiles (box), and min/max (whiskers). Data are from four independent experiments. *P < 0.05, **P < 0.01, ***P < 0.001, and ****P < 0.0001; t test. Source data are available for this figure.

First, we asked how secreted factors present in the breast cancer cell CM may lead to the activation of kinases observed in MSCs. To do this, we generated protein–protein interaction network models

of matrisome proteins enriched in the MDA-WT secretome versus the MCF-7 secretome, and kinases activated by the MDA-WT secretome (Fig 5A). We observed several interactions between

secreted proteins and kinases in MSCs, such as the interaction between IGFBP3 and the receptor IGF1R, and HSPG2 interacting with VEGFR2. These interactions could be targeted in future studies aiming to prevent TA-MSC activation. Similar network analyses for the MDA-Bone data are presented in Fig S6A. However, some ligands for activated kinases were not identified in our secretome analysis, which is likely because growth factors and cytokines are notoriously difficult to detect by mass spectrometry. Further fractionation and the use of new generation mass spectrometers might improve coverage and detection of small, low abundance proteins in secretome analyses.

When assessing the transcripts induced in MDA-WT–educated TA-MSCs, many of them are secreted molecules. These data suggest that TA-MSCs may secrete molecules that act in concert with the MDA-WT secretome to remodel their microenvironment. To provide an overview of potential new matrisome interactions in the TME upon TA-MSC activation, we generated protein–protein interaction network models of matrisome proteins enriched in the MDA-WT secretome and transcripts induced in MSCs by the MDA-WT secretome (Figs 5B and S6B). Fibronectin (FN1) is the most connected protein with 27 interaction partners, highlighting its potential role in coordinating MSC secretome functions. In support of the role of these proteins in human breast cancer, 11 matrisome proteins in the MDA-WT secretome (ANXA1, ANXA6, COL6A1, COL6A2, FN1, HSPG2, LGALS1, TGFBI, THBS1, THBS2, and TNC) and five proteins induced in MSCs (ALDOA, C3, COL3A1, GAPDH, and PGK1) have been identified with at least two unique peptides by proteomics in triple negative breast cancer patient samples previously (49) (CANX was induced in MSCs and was identified adjacent to the tumour tissue).

Network analysis points to an induced matrix remodelling in the TME. We tested whether such matrix remodelling may affect TA-MSC functions experimentally. We performed 3D spheroid-based invasion experiments by embedding MSC spheroids in PEG hydrogels, treating them with breast cancer secretomes and analysing their invasion (Fig 5C). Although MSCs strongly migrated into the surrounding hydrogel matrix when treated with the known chemo-attractant PDGF-BB, we observed no difference in MSC migration upon treatment with control medium, the MCF-7 secretome, or the MDA-WT secretome. This finding shows that breast cancer secretomes do not induce TA-MSC migration here, which indicates that the matrix-modulatory phenotype of TA-MSCs might be linked to nonmigratory matrix remodelling or to enhanced migration of other cells in the TME such as cancer cells or immune cells.

Finally, to functionally assess the immunomodulatory reprogramming of TA-MSCs identified in our transcriptome analysis, we carried out macrophage activation assays (Fig 5D). Macrophages are a further key component of the TME and have been shown to be recruited by TA-MSCs (41, 50, 51). To test whether TA-MSCs drive also the activation of macrophages, we generated naïve macrophages from the human monocyte THP-1 cell line as described previously (52) and treated them with either the breast cancer secretomes directly or with CM obtained from MSCs that were themselves pretreated with the breast cancer secretomes. After 24 h, we analysed the response of macrophages by qRT–PCR for established activation markers/cytokines (53). Strikingly, only the CM from MDA-WT–educated MSCs induced the expression of TNF, IL1B, and IL10, whereas secretomes from breast cancer cells or MSCs alone did not.

These data indicate that TA-MSCs induce features of both M1-like (TNF and IL1B) and M2-like (IL10) polarization (53).

Taken together, these data suggest that in response to the MDA-WT secretome, TA-MSCs do not increase their own migration capacity but are an important mediator between breast cancer cells and the activation of tumour-associated macrophages through secretion molecules. Future experiments are required to better understand the activation of macrophages by TA-MSCs, but the data provided here demonstrate an example of how our molecular profiling studies can be used to further investigate crosstalk between cells in the TME.

## Conclusions

Here, we have investigated MSC activation upon exposure to breast cancer secretomes from different stages of the metastatic cascade in 3D using PEG hydrogels. The great advantages of synthetic hydrogels are their low batch-to-batch variability, their fully defined and animal product free composition, and their precisely customizable biochemical and biophysical properties. These features enable the generation of reproducible and adaptable 3D cancer biology platforms, and future studies could decipher the effects of individual niche components of the TME such as matrix stiffness, specific ECM components, or cell-derived factors on TA-MSC activation. Our work on the molecular profiling of TA-MSC activation complements previous work using PEG hydrogels for the study of tumour morphogenesis, drug responses, and cell invasion (54, 55, 56, 57, 58, 59, 60, 61, 62). In addition, our omics datasets are complementary to other studies that have sought to identify metastasis-associated genes in breast cancer. In a recent study comparing the matrsiome of MDA-WT and MDA-Lung tumour xenografts (63), MMP1 and THBS2 were detected uniquely in MDA-WT tumours and AGRN was detected uniquely in MDA-Lung tumours, which correlates with our proteomics data of MDA-WT and MDA-Lung secretomes. Further tumour studies are required to assess the functional role of such metastasis-associated proteins.

This is the first time TA-MSC activation has been comprehensively investigated using invasive/noninvasive breast cancer cells and cells from different metastatic organs at the secretome, kinome, and transcriptome level. We identified ECM and exosome components associated with cancer progression and found that invasive MDA-MB-231 breast cancer cells activate TA-MSCs, whereas noninvasive MCF-7 cells do not. Further work is required to elucidate how proteins identified in this study contribute to the ability of invasive cancer cells to survive in new microenvironments during metastasis. As the organ-tropic breast cancer cells used here were selected through injection into immunocompromised mice, follow-up studies using immunocompetent mouse models of metastatic breast cancer to confirm the data presented here could be informative. In this manner, although we used patient-derived MSCs, it would also be interesting to confirm results using matched primary and metastatic cancer cells from patients. These data highlight breast cancer cell–derived proteins and those induced in the TME, specifically by MSCs, which warrant further study and could be targeted therapeutically to inhibit cancer progression.

# Materials and Methods

## Cell culture

Human bone marrow–derived MSCs were isolated as described previously ([64](ref)) from bone marrow aspirates of healthy donors (n = 4) obtained during orthopaedic surgical procedures after informed consent and in accordance with the local ethical committee (University Hospital Basel; Prof. Dr. Kummer; approval date 26/03/2007 Ref Number 78/07). MSCs were maintained in MEM $\alpha$ (with nucleosides; Gibco) supplemented with FBS (10%; Gibco), penicillin (100 U ml$^{-1}$; Gibco), streptomycin (100 $\mu$g ml$^{-1}$; Gibco), and FGF-2 (5 ng ml$^{-1}$; PeproTech) at 37°C in a humidified atmosphere at 5% $CO_2$.

We used human breast carcinoma cell lines that vary in their in vitro and in vivo invasiveness. Noninvasive MCF-7 cells were purchased from ATCC. Invasive MDA-MB-231 cells (MDA-WT) and MDA-MB-231 cells that were isolated from metastases in the bone, lung, and brain (MDA-Bone, MDA-Lung, and MDA-Brain) were kindly obtained from J Massagué at the Memorial Sloan-Kettering Cancer Center. Cancer cells were maintained in DMEM/Nutrient Mixture F-12 (DMEM/F12; Gibco) supplemented with FBS (10%; Gibco) and penicillin streptavidin (1%; Gibco) at 37°C in a humidified atmosphere at 5% $CO_2$. All cell lines were routinely tested for mycoplasma.

## Conditioned media preparation

CM was prepared as described previously ([65](ref)). Cancer cells (1 × 10$^6$) were seeded in 15-cm dishes in DMEM/F12 supplemented with 10% FBS. After 48 h, the medium was removed, the cells were washed twice with PBS, and 20 ml serum-free DMEM/F12 was added. For mass spectrometry analysis of CM, phenol red–free DMEM/F12 was used. CM was collected after 24 h, passed through a 0.22-$\mu$M filter, added to spin columns (Vivaspin 20 centrifugal concentrator with 10-kD molecular mass cutoff, Sartorius) and centrifuged (2,000$g$, 4°C) until the volume reached 1 ml. Concentrated 20× CM was snap-frozen and stored at –80°C until use. Concentrated 20× CM from three independent experiments was used for mass spectrometry analysis and 20× CM from three independent experiments was pooled for use with MSCs.

## Mass spectrometry acquisition of breast cancer cell secretomes

Concentrated CM in spin columns was diluted 1:2 in lysis buffer (6 M urea, 2 M thiourea, and 10 mM Hepes, pH 8) and centrifuged (2,000$g$, 4°C) until 500 $\mu$l volume remained. This was repeated three times in total. Dissolved proteins were heated (95°C, 5 min, 300 rpm [Biometra TSC Thermoshaker]) and sonicated at 4°C using a Bioruptor at high-energy setting (five cycles of 30 s on and 10 s off). Protein concentration was determined using the Bradford method. Denatured proteins were snap-frozen and stored at –80°C until required for mass spectrometry analysis.

For digestion, we took 50 $\mu$g of protein, which was diluted 1:3 with digestion buffer (50 mM Hepes, pH 8.5, 10% acetonitrile), and then digested first with LysC at a 1:50 (enzyme:protein) ratio for 3 h at 37°C. Subsequently, the samples were diluted to 1:10 (volume:volume) with digestion buffer, and digested overnight at 37°C. The following morning, the digests were quenched with 2% TFA to a final

concentration of 1%. Before MS analysis, 5 $\mu$g of each digest was desalted on in-house–packed StageTips ([66](ref)), dried down in an Eppendorf speed-vac, and resuspended in 10 $\mu$l of 1% TFA and 2% acetonitrile, containing Biognosys iRT peptides at a 1:1,000 dilution.

For MS analysis, from each sample, 1 $\mu$g of peptides were loaded onto a 2-cm C18 trap column (164705; Thermo Fisher Scientific), connected in-line to a 50-cm C18 reverse-phase analytical column (EasySpray ES803; Thermo Fisher Scientific) using 100% Buffer A (0.1% formic acid in water) at 750 bar, using the Thermo EasyLC 1000 HPLC system in a dual-column setup and the column oven operating at 45°C. Peptides were eluted over a 200-min gradient ranging from 5 to 38% of 80% acetonitrile and 0.1% formic acid at 250 nl/min, and the QExactive (Thermo Fisher Scientific) was run in a DD-MS2 manner. Full MS scans were collected at 70,000 resolution, with a 3 × 10$^6$ automatic gain control target and maximum injection time of 20 ms. MS2 scans were conducted at 17,500 resolution, with an automatic gain control target of 1 × 10$^6$ and 60-ms injection time. MS2 spectra were collected as a top 10 method, with a 1.6 m/z isolation window, 25 normalized collision energy, and a minimum intensity of 1.7 × 10$^4$. All unassigned and singly charged peptides were excluded. MS performance was verified for consistency by running complex cell lysate quality control standards, and chromatography was monitored to check for reproducibility. The raw mass spectrometry data have been deposited to the ProteomeXchange Consortium (http://proteomecentral. proteomexchange.org) via the PRIDE partner repository with the dataset identifier PXD010467.

Resulting .raw files were analysed using MaxQuant version 1.6.1.0 ([67](ref)) and standard settings. Briefly, label-free quantitation was enabled with a requirement of two unique peptides per protein, and iBAQ quantitation was also enabled during the search. Variable modifications were set as oxidation (M) and acetyl (protein N-term). Fixed modifications were set as carbamidomethyl (C), false discovery rate was set to 1%, and "match between runs" was enabled, with a 2-min alignment window.

In total, we identified 2,128 proteins from all biological replicates and cell lines. To identify proteins expressed uniquely in one secretome or another, we classified proteins as present in a secretome, provided label-free quantitation intensity values were greater than zero in at least two biological replicates, and we classified proteins as not expressed in a secretome if the intensity value was equal to zero across all three replicates, which are the criteria we have used previously ([68](ref)). To quantify differential abundance of proteins between samples, we filtered the dataset to include only those proteins identified in at least two replicates in all cancer cell secretomes analysed, and any missing values were imputed in the Perseus software package ([69](ref)), using the normal distribution with a width set to 0.3 and a down-shift of 1.8. This resulted in a high-confidence quantifiable list of 642 proteins (Table S1). Using this list, we performed multifactorial Limma analysis ([70](ref)) to determine significantly altered proteins ($P$ value < 0.05) between cancer cell secretomes. GO analysis was performed using DAVID (version 6.8) ([71](ref)) against the *Homo sapiens* background where terms with fold enrichment ≥1.5, Bonferroni-corrected $P$-value < 0.05, EASE score (modified Fisher's exact test) < 0.05, and at least two proteins per keyword were considered significantly overrepresented. ECM and secreted proteins were extracted based on annotations in the bioinformatics-based human Matrisome

database (http://matrisomeproject.mit.edu) (33). Exosome components were extracted based on the *H. sapiens* extracellular exosome GO annotation in AmiGO (version 2.5.12) (72).

## Western blotting of conditioned media

CM were mixed with 4× Laemmli sample buffer (containing 2-mercaptoethanol) and boiled for 10 min at 95°C. 15-$\mu$l protein volumes were separated on 4–15% Mini-PROTEAN TGX stain-free protein gels (4568086; Bio-Rad), and whole proteins were detected with the Criterion Stain-free imaging system (Bio-Rad). Next, the proteins were transferred on polyvinylidene difluoride membranes using the Trans-Blot-Turbo system (Bio-Rad). The membranes were blocked in 5% nonfat dry milk/TBS-T for 1 h at RT. The following primary antibodies were diluted 1:1,000 in 5% BSA/TBS-T and incubated overnight at 4°C: MMP1 (sc-30069; Santa Cruz), SERPINE1 (sc-5297; Santa Cruz), TIMP2 (ab53730; Abcam), COL6A1 (sc-377143; Santa Cruz), FN1 (F3648; Sigma-Aldrich), and THBS1 (sc-59887; Santa Cruz). Membranes were 3× washed in TBS-T and the following secondary antibodies were diluted 1:20,000 in 2.5% nonfat dry milk/TBS-T and incubated for 1 h at RT: goat-anti-mouse-HRP (SAB3701073-2; Sigma-Aldrich) or goat-anti-rabbit-HRP (SAB3700878-1; Sigma-Aldrich). HRP was visualized by the UltraScence Pico Ultra Western Substrate (CCH345-B; Gene-DireX) and the ChemiDoc MP imaging system (Bio-Rad).

## TG-PEG hydrogel formation and 3D cell culture

1 ml of FXIIIa (200 U ml$^{-1}$, Fibrogammin; CSL Behring) was activated with 100 $\mu$l of thrombin (20 U ml$^{-1}$; Sigma-Aldrich) for 30 min at 37°C. Small aliquots of FXIIIa were stored at –80°C for further use. Hydrogels with final dry mass contents of 1.7% wt/vol (corresponding to a storage modulus of 470 Pa (27)) were prepared by stoichiometrically balanced ([Lys]/[Gln] = 1) precursor solutions of n-PEG-Gln and n-PEG-MMP-sensitive-Lys (previously described (24, 25)) in 50 mM Tris buffer, pH 7.6, containing 50 mM CaCl$_2$. In addition, Lys-RGD at a final concentration of 50 $\mu$M and MSCs from four individual donors at a final concentration of 4 × 10$^6$ ml$^{-1}$ were added. Hydrogel volume was 50 $\mu$l. Subsequently, PEG cross-linking was initiated by addition of 10 U ml$^{-1}$ FXIIIa and disc-shaped matrices were prepared between hydrophobic glass slides (treated with SigmaCote; Sigma-Aldrich). Final hydrogels were cultured in a medium containing MEM $\alpha$, FBS (10%), penicillin (100 U ml$^{-1}$), and streptomycin (100 $\mu$g ml$^{-1}$) at 37°C in a humidified atmosphere at 5% CO$_2$. After 24 h, the medium was changed to minimal FBS conditions (2%). After 24 h, 50% medium was replaced with medium containing 1:10 diluted 20× CM. For the following 7 d, 50% medium was replaced every 24 h with medium containing 1:20 diluted 20× CM.

To retrieve cells, hydrogels were degraded in 0.5 mg ml$^{-1}$ collagenase A (11088793001; Sigma-Aldrich) at 37°C for 60 min. Subsequently, total RNA was isolated from cells using the RNeasy Micro Kit following the manufacturer's instructions (74004; QIAGEN). Alternatively, the cell pellets were snap-frozen and stored at –80°C until further use.

## Kinase activity profiling

To remove any collagenase A or PEG remaining from MSC retrieval, the cell pellets were washed in 100 $\mu$l cold PBS and centrifuged (1,000$g$, 4°C). The cells were lysed (10 min, 4°C, 500 rpm [Biometra TSC Thermoshaker]) in 25 $\mu$l Mammalian Protein Extraction Reagent (M-PER) (Thermo Fischer Scientific) containing EDTA-free protease (1×) and phosphatase (1×) inhibitor cocktails (Halt; Thermo Fischer Scientific). The cell lysates were clarified (10 min, 4°C, 16,000$g$) and protein concentration was determined using the Bradford method.

Kinase activity profiles were carried out using the PamChip Kinase Profiling Microarray System (PamGene) as described previously (37, 73). Separate chips for Protein Tyrosine Kinase (PTK) and the STK assays were used, where each chip contains peptides (with known phosphorylation sites) immobilised in an array format on a porous membrane. Membranes were first blocked with 2% BSA. 10 $\mu$l of cell lysate (2.7 and 1 $\mu$g in total for PTK and STK analyses, respectively) was mixed with 1× additive, 1× protein kinase (PK) buffer, 10 mM dithiothreitol, 1× BSA, 400 $\mu$M ATP, FITC-conjugated anti-phosphotyrosine antibody (PY20), and water to achieve a total volume of 40 $\mu$l. For the STK assay, a two-step procedure was applied involving a primary antibody and a second FITC-conjugated antibody. The samples were loaded onto the chip and were analysed using the Pamstation 12 instrument. Cell lysates were pumped through the membrane, allowing real-time phosphorylation of peptides by active kinases in the sample. Images were taken with a built-in CCD camera, and image analysis and signal quantification were performed using BioNavigator software (PamGene). Peptides that displayed higher signal than background were selected and pair-wise differences between conditions were analysed. Differences between the phosphorylation levels of individual peptides were calculated and were analysed using a *t* test (Tables S2 and S4). Next, data were analysed using the upstream kinase analysis app (2018 version; PamGene) with default settings, which allows prediction of kinases responsible for altered phosphorylation between conditions based on kinase–substrate relationships reported in multiple databases. The software gives as output the normalized kinase statistic (a proportional measure of activity for a kinase when evaluating a set of peptides that are linked to that kinase) and a combined score of significance (across replicates) and kinase specificity (indicates confidence in kinase prediction of a set of peptides to a kinase, so the higher the score, the less likely it is that the observed change in kinase activity could have been obtained using a random set of peptides) (Tables S3 and S5).

## RNA sequencing

RNA concentration was determined with RiboGreen (Life Technologies) and measured on the Infinite M1000 Pro plate reader (Tecan). RNA library preparation was performed using a polyA selection method. RNA sequencing was performed using the Illumina HiSeq system in a 2 × 150-bp configuration (single index, per lane) by GENEWIZ and RNAseq data were analysed on the Galaxy server at BRIC, University of Copenhagen (https://bricweb.sund.ku.dk/galaxy/) (74). Starting from the raw .fastq files, the reads were mapped against the human reference genome (hg19Full) using the RNA STAR aligner (version 2.4.0d-2) (75). Read counting was performed with htseq-count (version 0.6.0) (76) using union mode. Differential gene expression was analysed using DESeq2 (version 1.8.2) (77). Significance was defined as *P* value < 0.05 after adjustment for multiple testing with the Benjamini–Hochberg procedure.

Functional enrichment maps were generated as described previously ([78], [79]). Overrepresentation of GO terms was calculated for the list of significantly altered genes using High-Throughput GoMiner ([80]) using the *H. sapiens* background. One thousand randomisations were performed, and data were thresholded for a 0.05 false discovery rate. Overrepresented terms with ≥5 and ≤500 assigned genes were reported. Fold change values were mapped onto genes assigned to each overrepresented term.

### Hierarchical clustering

Hierarchical clustering analysis was performed on the basis of uncentred Pearson correlation and a complete-linkage matrix using Cluster 3.0 ([81]). Clustered data were visualized using Java TreeView ([82]).

### qRT–PCR

MSCs from four individual donors were 3D-cultured, and RNA was isolated as described above. 200 ng RNA were converted into 60 $\mu$l cDNA by means of the High-Capacity cDNA Reverse Transcription Kit (Applied Biosystems). qRT-PCR was carried out using 1.5 $\mu$l cDNA template, the TaqMan Universal PCR Master Mix (Applied Biosystems), and the StepOnePlus Real-Time PCR System (Applied Biosystems). The following TaqMan primer/probe sets were used for gene expression tests: Hs00236937_m1 (CXCL1), Hs00601975_m1 (CXCL2), Hs00171061_m1 (CXCL3), Hs01099660_g1 (CXCL5), Hs00174103_m1 (CXCL8), Hs00899658_m1 (MMP1), Hs00968305_m1 (MMP3), Hs01029057_m1 (MMP8), Hs00957562_m1 (MMP9), Hs00942584_m1 (MMP13), Hs00737347_m1 (PFKP), Hs01378790_g1 (LDHA), Hs00299262_m1 (SLC39A14), and Hs01061804_g1 (SLC39A8). Data were normalized on the expression of the following genes: Hs03044281_g1 (YWHAZ) and Hs02800695_m1 (HPRT1). Relative gene expression was calculated by the comparative Ct method.

### Network analysis

Interaction network analysis was performed using Genemania (version 3.4.1) ([83]) within Cytoscape (version 3.4.0) ([84]) where enriched proteins were mapped onto a human interactome consisting of reported physical protein–protein interactions. Unconnected proteins were manually removed.

### Macrophage activation

The human monocyte THP-1 cell line was obtained from the American Type Culture Collection and cultured in RPMI (Sigma-Aldrich) supplemented with 10% FBS and penicillin/streptomycin at 37°C in a humified 5% $CO_2$ atmosphere. Differentiation of monocytes (0.35 × 10$^6$, 12-well) towards naïve macrophages was induced by stimulating with 100 nM phorbol 12-myristate 13-acetate (PMA; Sigma-Aldrich) in full RPMI medium for 3 d, followed by 24 h recovery in full RPMI medium. Next, naïve macrophages were cultured in minimal MEM $\alpha$ control medium (2% FBS) or supplemented with 1:20 diluted 20× CM from MCF-7 or MDA-MB-231. Alternatively, macrophages were cultured in CM collected from MSCs (N = 4) that were treated with control medium or CM from MCF-7 or MDA-MB-231

as described above (cultured in hydrogels for 7 d with daily 50% medium changing). More specifically, the 50% MSC CM was collected daily and pooled over the 7-d time course. 2 ml of prepared medium was added to macrophages, and RNA was isolated after 24 h of culture. 500 ng RNA were converted into 30 $\mu$l cDNA and qRT–PCR was performed with the following TaqMan primer/probe sets: Hs00961622_m1 (IL10), Hs00174097_m1 (IL1B), Hs01113624_g1 (TNF), and Hs00998133_m1 (TGFB1). Data were normalized to the expression of the following genes: Hs02758991_g1 (GAPDH) and Hs01060665_g1 (ACTB). Relative gene expression was calculated by the comparative Ct method.

### 3D spheroid migration assay

For spheroid formation, MSC cell suspensions (final. conc. 3.3 × 10$^5$ cells ml$^{-1}$) were suspended in MEM $\alpha$ (2% FCS) and supplemented with 0.2% (wt/vol) methyl cellulose (Cat. No. M0512; Sigma-Aldrich). Droplets of 30 $\mu$l were placed in nonadhesive cell culture dishes and cultured for 24 h as hanging drops. The resulting spheroids (1,000 cells) were harvested in MEM $\alpha$ (2% FCS) and washed once with medium.

MSC spheroids were encapsulated in PEG hydrogels (1.7% wt/vol, as described above) and cultured in MEM $\alpha$ (2% FCS) supplemented with 1:20 diluted 20× CM; with a daily replacement of 50% medium containing 1:20 diluted 20× CM. As positive control, we used 50 ng ml$^{-1}$ human recombinant PDGF-BB (Peprotech 100-14B). After 72 h, the hydrogels were washed twice with PBS followed by fixation in 4% PFA for 30 min at RT. Next, the gels were washed for 1 h with PBS and stained for F-actin with rhodamine phalloidin (Molecular Probes 1:500) in PBS containing 1% BSA (Albumin Fraction V, AppliChem) for 4 h at RT.

Z-stack images covering the whole spheroid were acquired by an inverted laser scanning confocal microscope (TCS SP5; Leica). The images were Z-projected, binarized, and automatically batch-measured by the *analyze particles* tool of ImageJ software (Fiji version 1.48.u, April 2014) for the whole area covered by cells (summed F-actin–positive signal of pixels >10 $\mu m^2$). 3D migration was calculated by subtracting the area of a perfectly circular initial spheroid from the measured whole area. 12 spheroids from three independent experiments (using three different MSC donors) were measured per condition.

## Data Availability

Proteomics data have been deposited in ProteomeXchange (http://proteomecentral.proteomexchange.org) through the PRIDE partner repository ([85]) with the primary accession identifier PXD010467. RNA sequencing data have been deposited to the Annotare database (https://www.ebi.ac.uk/fg/annotare) with the identifier E-MTAB-6998.

## Supplementary Information

# Acknowledgements

We would like to thank GENEWIZ (UK) for acquisition of RNA sequencing data and Pamgene (The Netherlands) for technical and data analysis advice regarding kinase activity profiling. We are very thankful to Queralt Vallmajo-Martin (University Hospital Zurich) for PEG functionalisation. We further thank Maja Bollhalder (Balgrist University Hospital) for giving advice on Western blot experiments. This work was supported by the European Research Council (ERC-2015-CoG-682881-MATRICAN), the European Molecular Biology Organization (ALTF 922-2016), the Danish Cancer Society (R204-A12445), the European Union's Seventh Framework Programme (iTERM grant agreement No. 607868) and the Swiss National Science Foundation (310030-169808/1).

## Author Contributions

U Blache: conceptualization, data curation, formal analysis, investigation, visualization, and writing—original draft, review, and editing.

ER Horton: conceptualization, data curation, formal analysis, funding acquisition, investigation, visualization, and writing—original draft, review, and editing.

T Xia: formal analysis and methodology.

EM Schoof: formal analysis and methodology.

LH Blicher: formal analysis and methodology.

A Schönenberger: methodology

JG Snedeker: supervision and funding acquisition

I Martin: resources and supervision

JT Erler: conceptualization, supervision, funding acquisition, and writing—review and editing.

M Ehrbar: conceptualization, supervision, funding acquisition, and writing—review and editing.

## Conflict of Interest Statement

The authors declare that they have no conflict of interest.

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
