## [Reviewer comments · Life Science Alliance]

Life Science Alliance

Mesenchymal stromal cell activation by breast cancer secretomes in synthetic 3D hydrogels

Ulrich Blache, Edward Horton, Tian Xia, Erwin Schoof, Lene Blicher, Angelina Schönenberger, Jess Snedeker, Ivan Martin, Janine Erler, and Martin Ehrbar

DOI: <https://doi.org/10.26508/lsa.201900304>

Corresponding author(s): Martin Ehrbar, University of Zurich and Janine Erler, University of Copenhagen

Review Timeline:	Submission Date:	2019-01-14
	Editorial Decision:	2019-02-12
	Revision Received:	2019-05-10
	Editorial Decision:	2019-05-21
	Revision Received:	2019-05-22
	Accepted:	2019-05-22

Scientific Editor: Andrea Leibfried

Transaction Report:

February 12, 2019

Re: Life Science Alliance manuscript #LSA-2019-00304-T

Dr. Martin Ehrbar
University of Zurich
Obstetrics
Schmelzbergstrasse 12
Zurich, Zurich 8091
Switzerland

Dear Dr. Ehrbar,

Thank you for submitting your manuscript entitled "Mesenchymal stromal cell activation by breast cancer secretomes in synthetic 3D hydrogels" to Life Science Alliance. The manuscript was assessed by expert reviewers, whose comments are appended to this letter.

As you will see, the reviewers think that further validation for the proteins/kinases/genes identified here is needed to make sure that your manuscript provides a valuable resource to others. They furthermore note that the different parts of the manuscript are currently ill-connected. Should you be able to address these major concerns in a satisfactory way, we'd be happy to consider a revised version of your manuscript further here.

To submit a revised version of your manuscript, please log in to your account:

Thank you for this interesting contribution to Life Science Alliance. We are looking forward to receiving your revised manuscript.

Sincerely,

Andrea Leibfried, PhD
Executive Editor
Life Science Alliance
Meyershofstr. 1
69117 Heidelberg, Germany
t +49 6221 8891 502
e a.leibfried@life-science-alliance.org
www.life-science-alliance.org

B. MANUSCRIPT ORGANIZATION AND FORMATTING:

Reviewer #1 (Comments to the Authors (Required)):

Blache and colleagues have investigated mechanisms by which breast cancer cells from different stages of the metastatic cascade convert MSCs into tumour-associated MSCs (TA-MSCs) using global approaches. They have relied on gene expression analyses to determine gene sets that are

upregulated and down regulated when compared between secretomes from invasive and non-invasive breast cancer cells. They have also examined differential ability to convert MSCs to TA-MSCs following exposure of MSCs in 3D hydrogel matrices to conditioned media from invasive and non-invasive breast cancer cells. They further report that only invasive breast cancer secretomes convert MSCs into TA-MSCs, resulting in an immune-modulatory phenotype that was particularly prominent in response to bone-tropic cancer cells. Several interesting gene sets were identified that may be involved in determining both the invasive phenotype as well as the gene sets associated with TA-MSCs.

The rationale of the work was to better understand how MSCs are converted to TA-MSCs.

It appears that they have made some progress in this regard however the specific mechanisms associated with the conversion process remains elusive.

The authors provide very general conclusions and most of the gene sets or pathways appear to be similar to previously identified players. No new insight is offered regarding selection of druggable targets.

No validation of identified proteins/gene sets is provided.

Reviewer #2 (Comments to the Authors (Required)):

The authors provide several datasets detailing the effects of the conditioned media of various commercially available and previously established human breast non-invasive and metastatic cancer cell lines on primary mesenchymal stem cells. In particular they have performed a mid-depth MS-proteomic characterisation of the secretome of the cancer cell lines and a kinome (using a kit) and RNA-seq analysis of mesenchymal stem cells cultured in PEG and treated with the conditioned media from the different cancer cell lines. For each omic analysis, the authors provide some statistical and bioinformatic analysis and make some biological conclusions/speculations based on the results. The secretomes of some of the cell lines (MDA-WT and MDA-Bone) have been published before by the Eler's group (Cox et al. Nature 2015), however, here the study is extended to additional lines that metastasise to different organs. Furthermore, the effects of breast cancer cell secretome on patient-derived mesenchymal stromal cells has not been studied in depth and therefore represents a novel and useful resource for the cancer field.

Generally the experiments performed and the various omics analysed are of good quality and they would be a resource of interest to the cancer field however there are concerns with some of the conclusions. The authors need to provide more evidence to back up their statements or tone-down some of them before the manuscript is suitable for publication. Moreover, some further analysis would improve the impact of the work.

While I did find the overall approach and models very interesting, I was a bit disappointed by the fact that I could not find a connection between the different omic analyses. First, the authors analyse the secretome of the different cancer cell lines and found differentially secreted ECM components and extracellular vesicle-related proteins. Then they perform a kinome analysis of the mesenchymal stem cells upon treatment with cancer cell-derived conditioned media, which highlights several regulated TKR and other kinases. However, in the manuscript there is no mention of soluble factors, such as growth factors, cytokines and chemokines that were found in the conditioned media and

that could be ligands of and activate those receptors. Where any of those ligands detected? If there were no soluble factors in the secretomes, why is that (maybe the limited depth of the study? This is actually suggested by the fact that some proteins identified in this study were not identified in the previous study, Nature 2015, and vice versa)? The authors should mention possible limitations of their analyses. Next, the authors move on with performing RNAseq of the mesenchymal stem cells following treatment with cancer cell-derived conditioned media. Also for these results, the authors do not provide any connection with the kinome analysis, nor with secreted factors that could trigger such gene expression activation. An analysis of the RNAseq data to predict possible regulated transcription factors could be informative. Moreover, Could the authors use some modelling to link the different analyses with each other and model the response of the different cell types? I do understand that this may require quite some work, but it would really help to better interpret the different models. Maybe this could explain why the kinome of the metastatic lines is different, but not their transcriptome?

The metastatic MDA-MB-231 lines used by the authors are an excellent tool to discerning the effects of clinically relevant metastatic sites on stromal compartments compared to the primary site - this is one of the main strengths of the paper. However, these lines have been generated through in vivo selection of xenografts into immunocompromised balb/c nude mice. The authors should comment on how these lines may have evolved less immunomodulating characteristics than metastases found in human patients, and this in turn may affect their relevance to MSC activation, especially when changes in key cyto/chemokines are a conclusion of the authors. An excellent follow up study would be to isolate cancer cells from human patients at the primary and metastatic sites, or alternatively use an immunocompetent mouse model of metastatic breast cancer to confirm their findings, although this may be beyond the scope of the current study.

To strengthen the validity of the datasets provided by the authors as a resource for the community, they may include validation of some of the regulated proteins/kinases/genes. They could validate at least one target from each omics performed. For example MMP1, which is major finding and which could be easily validated by western blot or gel zymography; VEGFR2 which has well-characterised phosphorylation specific antibodies for western blotting.

Page 8. The first two sentences are not clear. It seems that the authors did not find any difference when cells were treated with MDA-WT conditioned media or serum-free media.

Supplementary Methods, Mass spectrometry acquisition...: Have the authors used any threshold of significance for the LIMMA analysis?

Reviewer #3 (Comments to the Authors (Required)):

Blache and Horton et al. studied the paracrine impact of breast cancer cells with different invasive and metastatic behaviour on the differentiation of MSCs from 4 human donors performing secretome, kinase activity and RNAseq profiling. MSCs were grown in biomimetic PEG hydrogels and treated with the conditioned medium of the different breast cancer cells for 7 days. The breast cancer secretomes and MSC kinome and transcriptome were analysed. While the secretome and kinase as well as the breast cancer cell transcriptomic experiments produced significant data, the transcriptomic experiments of MSCs didn't show any significant changes.

This is an interdisciplinary study between bioengineers, tumour biologists and bioinformaticians,

highlighting the importance of tissue engineering tools that can be applied to cancer research. This controllable and reproducible approach helps scientists to decipher the roles of the individual matrix and cellular components of tumour microenvironment in cancer progression.

Validation experiments using kinase-inhibitory compounds and invasion assays would strengthen this study. The role of some of the identified factors in the bone, lung or brain tumour microenvironment should be discussed or even compared with publically available transcriptomic or proteomic datasets and reported breast cancer studies as similar factors were identified previously. The conclusions need to be revised.

Major comments:

- 1.The authors should consider changing the title to 'Mesenchymal stromal cell activation by breast cancer secretomes in bioengineered 3D microenvironments'.
- 2.Page 5, results, which specific matrixome database do the authors refer to? For primary or metastatic breast cancer? The numbers for the 68 and 64 differentially expressed exosome/matrixome proteins in MDA-bone versus MDA-lung should be included in figure 1F. At the end of this paragraph, the discussion on why the bone/lung-metastatic cells had a different secretome compared to the brain-metastatic cells should be extended? What is difference in their microenvironment? What about the blood-brain barrier?
- 3.Page 6, results, what makes PEG hydrogels biomimetic? Explain this here in more detail again (or establish the term 'biomimetic' in the introduction) and why this is important for MSCs. Why was perlecan, a large proteoglycan found in the vascular ECM, chosen as representative ECM factor in figure 2A? Is the medium for the secretome cultures different to the actual MSC culture medium? Was the secretome medium replaced during the 7-day treatment period? Why was a 7-day treatment chosen? Transcriptomic changes will occur much earlier. The secretome analysis was performed after 24 hours using serum-free conditions, while the effect of the conditioned medium on MSCs was measured after 7 days.
- 4.Page 7, results, a PTK inhibitor or specific blocking antibody could be used to validate the changes in the kinase profile in MSCs. Include references for statement 'are known to be involved in cancer and angiogenesis'. The advantage of RNAseq over MS/protein analysis should be better explained as this technique also helps to identify novel transcripts and genes that might not encode for functional proteins.
- 5.Pages 9-10, results, include references for statement 'aerobic glycolytic activity is a hallmark of cancer and is known as the Warburg effect'. Why is COMP an important factor in bone or lung metastasis?
- 6.The conclusions need to be revised and should include some therapeutics that are important for targeting TME components that promote breast cancer and metastasis. What is the stiffness of MSC-seeded hydrogels? Does it change with treatment using the different secretomes? PEG hydrogels are an established model for 3D cultures and disease platforms as well as multi-omics and multi-level analyses. Different PEG matrices are used by various different researchers around the world and more cancer-specific references should be included for people readers which are not familiar with these hydrogels.

Minor comments:

- 1.In the abstract, 'bioengineered synthetic' should rather read 'synthetic'; use at the end of this sentence 'bioengineered 3D microenvironment'; delete 'PEG'. What specific anti-cancer therapeutics target cancer-supporting cells? What about anti-metastatic treatments? The last sentence should be extended.
- 2.Presumably, the MDA-bone/lung/brain cells are human. How were they derived? The authors should also refer to the original paper in which these cell lines have been established.

3. Page 3, introduction, use the abbreviation 'TME' for tumour microenvironment. What are 'confounding signals'? Use 'cytocompatible' instead of 'cell-friendly'. Explain abbreviations upon first usage, for example MMP.
4. Pages 7-9, results, reword 'milder' and 'mild'.
5. Legends for figures 1 and 2 need to be shortened.
6. Page numbers in the supplementary methods are missing.

Reviewer #1 (Comments to the Authors (Required)):

Blache and colleagues have investigated mechanisms by which breast cancer cells from different stages of the metastatic cascade convert MSCs into tumour-associated MSCs (TA-MSCs) using global approaches. They have relied on gene expression analyses to determine gene sets that are upregulated and down regulated when compared between secretomes from invasive and non-invasive breast cancer cells. They have also examined differential ability to convert MSCs to TA-MSCs following exposure of MSCs in 3D hydrogel matrices to conditioned media from invasive and non-invasive breast cancer cells. They further report that only invasive breast cancer secretomes convert MSCs into TA-MSCs, resulting in an immune-modulatory phenotype that was particularly prominent in response to bone-tropic cancer cells. Several interesting gene sets were identified that may be involved in determining both the invasive phenotype as well as the gene sets associated with TA-MSCs.

The rationale of the work was to better understand how MSCs are converted to TA-MSCs.

It appears that they have made some progress in this regard however the specific mechanisms associated with the conversion process remains elusive.

The authors provide very general conclusions and most of the gene sets or pathways appear to be similar to previously identified players. No new insight is offered regarding selection of druggable targets.

No validation of identified proteins/gene sets is provided.

Response: We thank the reviewer for assessing our manuscript and for his/her feedback. We agree with this reviewer that some of the gene sets we have identified are similar to previously identified players, such as for example the cytokines induced in MSCs by the cancer cell secretomes. We have provided references in the text to state that some of these induced proteins are already known. However, we believe that identifying previously known proteins strengthens our study and acts as a validation of our dataset.

To further validate our omics datasets and strengthen the quality of our research resource for the scientific community, we have performed additional experiments and analyses as suggested by the reviewer.

Cancer cell secretomes: We have performed western blots where we have probed the 5 different cancer cell secretomes for 6 proteins that we identified to be differentially changed in abundance between secretomes (**Figure 1F**). These data show that different proteins are present at different abundances across the secretomes, and these data correlate with the mass spectrometry results. For example, THBS1 showed highest abundance in the MDA-Bone secretome.

Kinases: Our kinase activity profiling data are based on a prediction tool, which allows prediction of kinases responsible for altered phosphorylation between conditions based on

kinase-substrate relationships reported in multiple databases. Using this system, we are able to measure the phosphorylation of peptides from our cell lysate in real time using small amounts of material (2.7 μ g for tyrosine kinases and 1 μ g serine/threonine kinases), which is advantageous when using cell lysates extracted from 3D hydrogels as we have done here. We attempted to extract MSCs from hydrogels after incubation with cancer cell secretomes and perform western blots using phospho-specific antibodies, but unfortunately we were not able to collect enough material to detect phosphorylation signals. Instead, we have performed an additional analysis of our existing data, where we have compared the intensity levels of specific phosphopeptides between conditions. These analyses form part of **Figures 2C, S3A and S4 and supplementary tables 2 and 4**.

RNAseq: We have performed qRT-PCR analysis for 14 genes that were either induced in MSCs in response to the MDA-WT secretome compared with the MCF-7 secretome (**Figure S5A**), or were induced in MSCs in response to the MDA-Bone secretome (**Figure S5B**).

We are aware that future studies are needed to address the specific mechanism of TA-MSC conversion. However, a comprehensive picture of how cancer cell secretomes activate MSCs is currently lacking, since current knowledge is based on scattered observations from different studies that use different experimental set ups (eg. 2D, 3D, in vitro, in vivo) and different cancer cell types (some mouse, some human and from different cancer types). Our data in contrast to many previous studies are generated using human cells in a 3D in vitro model of the TME, therefore providing a global picture using a well-defined 3D system. Also, MSC responses to secretomes isolated from cancer cells from different metastatic organs has, to our knowledge, never been performed. In this regard, we believe that our manuscript will be a very helpful and valuable research resource to future studies.

We have toned down our conclusions by making changes to the text throughout the manuscript. Instead of referring to invasive and non-invasive cells, we have referred to MDA-WT and MCF-7 cells instead to be more explicit or deleted reference to invasion altogether. We have also removed 'invasion-associated signature' and have added some text highlighting limitations of using cancer cells from an immunocompromised model (conclusions section).

Reviewer #2 (Comments to the Authors (Required)):

The authors provide several datasets detailing the effects of the conditioned media of various commercially available and previously established human breast non-invasive and metastatic cancer cell lines on primary mesenchymal stem cells. In particular they have performed a mid-depth MS-proteomic characterisation of the secretome of the cancer cell lines and a kinome (using a kit) and RNA-seq analysis of mesenchymal stem cells cultured in PEG and treated with the conditioned media from the different cancer cell lines. For each omic analysis, the authors provide some statistical and bioinformatic analysis and make some biological conclusions/speculations based on the results. The secretomes of some of the cell lines (MDA-WT and MDA-Bone) have been published before by the Erler's group (Cox et al. Nature 2015), however, here the study is extended to additional lines that metastasise to

different organs. Furthermore, the effects of breast cancer cell secretome on patient-derived mesenchymal stromal cells has not been studied in depth and therefore represents a novel and useful resource for the cancer field.

Generally the experiments performed and the various omics analysed are of good quality and they would be a resource of interest to the cancer field however there are concerns with some of the conclusions. The authors need to provide more evidence to back up their statements or tone-down some of them before the manuscript is suitable for publication. Moreover, some further analysis would improve the impact of the work.

Response: We thank the reviewer for assessing our manuscript and for his/her overall encouraging and positive verdict. Based on his/her constructive comments, we have added further experimental data and revised the manuscript as detailed below.

While I did find the overall approach and models very interesting, I was a bit disappointed by the fact that I could not find a connection between the different omic analyses. First, the authors analyse the secretome of the different cancer cell lines and found differentially secreted ECM components and extracellular vesicle-related proteins. Then they perform a kinome analysis of the mesenchymal stem cells upon treatment with cancer cell-derived conditioned media, which highlights several regulated TKR and other kinases. However, in the manuscript there is no mention of soluble factors, such as growth factors, cytokines and chemokines that were found in the conditioned media and that could be ligands of and activate those receptors. Where any of those ligands detected? If there were no soluble factors in the secretomes, why is that (maybe the limited depth of the study? This is actually suggested by the fact that some proteins identified in this study were not identified in the previous study, Nature 2015, and vice versa)? The authors should mention possible limitations of their analyses.

Response: We thank the reviewer for raising these points about our mass spectrometry data of cancer cell secretomes. When we analysed our data, we used stringent criteria in the analysis. Proteins must have been identified with at least 2 unique peptides in at least 2 replicates in all secretomes analysed, which resulted in a high-confidence quantifiable list of 642 proteins used in our analysis. As growth factors, cytokines and chemokines are notoriously difficult to detect by mass spectrometry, these criteria likely meant that we have missed some of these proteins from our analysis, particularly those that are present in one secretome and not another. We have mentioned this point in page 13 of the manuscript. Therefore, we have performed additional analyses of our data. We chose to classify proteins as present in a secretome provided LFQ intensity values were greater than zero in at least two biological replicates, and we classified proteins as not expressed in a secretome if the intensity value was equal to zero across all three replicates, which are criteria we have used previously (Xia et al 2018, Proteomics). Using this approach, we identified additional proteins that are specific to certain cancer cell secretomes and are presented in **Figure S2**. These additional proteins include secreted factors, such as TGF β 1, BMP4, CSF1 and PDGF.

Next, the authors move on with performing RNAseq of the mesenchymal stem cells following treatment with cancer cell-derived conditioned media. Also for these results, the authors do

not provide any connection with the kinome analysis, nor with secreted factors that could trigger such gene expression activation. An analysis of the RNAseq data to predict possible regulated transcription factors could be informative.

Response: We attempted to analyse our RNAseq data using BART (binding analysis for regulation of transcription) (Wang et al 2018, Bioinformatics). The output we got was a long list of transcription factors that may regulate the changes in gene expression that we see. We feel that this analysis actually adds a long list of proteins to our manuscript, confusing the reader, so we have decided to not include it in the revised version.

Moreover, Could the authors use some modelling to link the different analyses with each other and model the response of the different cell types? I do understand that this may require quite some work, but it would really help to better interpret the different models. Maybe this could explain why the kinome of the metastatic lines is different, but not their transcriptome?

Response: We agree with this reviewer that it could be informative to connect the different omics analyses that we have performed. We have not attempted to link the kinases with the gene expression changes, since these data were analysed at the same time point so it is not possible to distinguish between cause and effect. Rather, the kinase activity and RNAseq data form two different readouts of MSC responses. We believe that the most useful approach is to connect the proteins identified in the conditioned media with the kinase data, since the secretomes are what trigger the kinase responses. As indicated by the reviewer, while a comprehensive modeling approach could be useful, it would be a lot of work and we feel is beyond the scope of our manuscript. However, we have performed interaction network analyses to highlight possible interactions between ligands in the cancer cell conditioned media and kinases present in MSCs. In addition, we have provided networks of proteins in the cancer cell conditioned media and transcripts induced by MSCs to provide an overview of potential new matrix proteins in the tumour microenvironment upon MSC activation, since many induced transcripts were secretion molecules. These networks form **Figure 5A,B and S6** and can be used to generate hypotheses of how TA-MSC activation in response to cancer cell secretomes occurs. We have also added text to page 11 and 12.

The metastatic MDA-MB-231 lines used by the authors are an excellent tool to discerning the effects of clinically relevant metastatic sites on stromal compartments compared to the primary site - this is one of the main strengths of the paper. However, these lines have been generated through in vivo selection of xenografts into immunocompromised balb/c nude mice. The authors should comment on how these lines may have evolved less immunomodulating characteristics than metastases found in human patients, and this in turn may affect their relevance to MSC activation, especially when changes in key cyto/chemokines are a conclusion of the authors. An excellent follow up study would be to isolate cancer cells from human patients at the primary and metastatic sites, or alternatively use an immunocompetent mouse model of metastatic breast cancer to confirm their findings, although this may be beyond the scope of the current study.

Response: We agree with the reviewer that there may be limitations to using cells from an immunocompromised model. As such we have mentioned that using an immunocompetent mouse model and patient cells would be excellent follow up studies in page 15.

To strengthen the validity of the datasets provided by the authors as a resource for the community, they may include validation of some of the regulated proteins/kinases/genes. They could validate at least one target from each omics performed. For example MMP1, which is major finding and which could be easily validated by western blot or gel zymography; VEGFR2 which has well-characterised phosphorylation specific antibodies for western blotting.

Response: We thank the reviewer for the comment and agree. We have followed the suggestion and performed targeted validation experiments and new analyses as detailed here:

Cancer cell secretomes: We have performed western blots where we have probed the 5 different cancer cell secretomes for 6 proteins that we identified to be differentially changed in abundance between secretomes (**Figure 1F**). These data show that different proteins are present at different abundances across the secretomes, and these data correlate with the mass spectrometry results. For example, THBS1 showed highest abundance in the MDA-Bone secretome.

Kinases: Our kinase activity profiling data are based on a prediction tool, which allows prediction of kinases responsible for altered phosphorylation between conditions based on kinase-substrate relationships reported in multiple databases. Using this system, we are able to measure the phosphorylation of peptides from our cell lysate in real time using small amounts of material (2.7 μ g for tyrosine kinases and 1 μ g serine/threonine kinases), which is advantageous when using cell lysates extracted from 3D hydrogels as we have done here. We attempted to extract MSCs from hydrogels after incubation with cancer cell secretomes and perform western blots using phospho-specific antibodies, but unfortunately we were not able to collect enough material to detect phosphorylation signals. Instead, we have performed an additional analysis of our existing data, where we have compared the intensity levels of specific phosphopeptides between conditions. These analyses form part of **Figures 2C, S3A and S4 and supplementary tables 2 and 4**.

RNAseq: We have performed qRT-PCR analysis for 14 genes that were either induced in MSCs in response to the MDA-WT secretome compared with the MCF-7 secretome (**Figure S5A**), or were induced in MSCs in response to the MDA-Bone secretome (**Figure S5B**).

Page 8. The first two sentences are not clear. It seems that the authors did not find any difference when cells were treated with MDA-WT conditioned media or serum-free media.

Response: The point we are trying to make is that there were no differences between serum-free versus MCF-7, and therefore when comparing each of these to MDA-WT we see largely the same outcome. We have rephrased these sentences to make this clearer, now on page 9.

Supplementary Methods, Mass spectrometry acquisition...: Have the authors used any threshold of significance for the LIMMA analysis?

Response: Yes, we have added $P < 0.05$ to the LIMMA methods on page 18.

Reviewer #3 (Comments to the Authors (Required)):

Blache and Horton et al. studied the paracrine impact of breast cancer cells with different invasive and metastatic behaviour on the differentiation of MSCs from 4 human donors performing secretome, kinase activity and RNAseq profiling. MSCs were grown in biomimetic PEG hydrogels and treated with the conditioned medium of the different breast cancer cells for 7 days. The breast cancer secretomes and MSC kinome and transcriptome were analysed. While the secretome and kinase as well as the breast cancer cell transcriptomic experiments produced significant data, the transcriptomic experiments of MSCs didn't show any significant changes.

This is an interdisciplinary study between bioengineers, tumour biologists and bioinformaticians, highlighting the importance of tissue engineering tools that can be applied to cancer research. This controllable and reproducible approach helps scientists to decipher the roles of the individual matrix and cellular components of tumour microenvironment in cancer progression.

Validation experiments using kinase-inhibitory compounds and invasion assays would strengthen this study. The role of some of the identified factors in the bone, lung or brain tumour microenvironment should be discussed or even compared with publically available transcriptomic or proteomic datasets and reported breast cancer studies as similar factors were identified previously. The conclusions need to be revised.

Response: We thank the reviewer for assessing our manuscript and for his/her positive feedback and for the suggestions to improve the quality of the manuscript. Based on his/her constructive comments and suggestions, we have revised the manuscript as detailed below.

As suggested by the reviewer, we have performed invasion assays of MSCs treated with control, MCF-7 and MDA-WT secretomes (**Figure 5C**). As we did not see any differences in invasion between these conditions, we did not test any inhibitory compounds in this assay. There are several possible explanations for why the secretomes did not lead to differences in invasion, which we have mentioned in the text on page 13. One hypothesis is that TA-MSc matrix remodeling caused by the increase in MMPs observed in MSCs treated with the MDA-WT secretome does not affect MSC migration, but that of other cells. Therefore, we chose to perform an additional functional assay related to the immune-modulatory phenotype we saw in the RNAseq data. We treated macrophages with several secretomes (**Figure 5D**) and assessed macrophage activation markers by qRT-PCR. We found that the only secretome that induces these markers is the MSC secretome collected from MSCs pre-treated with the MDA-WT secretome.

Also, we have performed additional validation assays as indicated in the response to Reviewer 1.

We have compared our proteomics data to two studies that have used ECM proteomics to study the matrisome in breast cancer. We have added sentences mentioning these comparisons to page 12 and 14. We have also added references referring to COMP in metastasis to page 11 and have mentioned the observed reverse Warburg effect we see in response to the MDA-Bone secretome on page 10. We have also added some discussion regarding the blood-brain barrier and possible reasons why we do not see gross changes to the MDA-Brain secretome compared to MDA-WT to page 6.

Major comments:

1. The authors should consider changing the title to 'Mesenchymal stromal cell activation by breast cancer secretomes in bioengineered 3D microenvironments'.

Response: We agree to this suggestion and we have changed the title.

2. Page 5, results, which specific matrisome database do the authors refer to? For primary or metastatic breast cancer? The numbers for the 68 and 64 differentially expressed exosome/matrisome proteins in MDA-bone versus MDA-lung should be included in figure 1F. At the end of this paragraph, the discussion on why the bone/lung-metastatic cells had a different secretome compared to the brain-metastatic cells should be extended? What is difference in their microenvironment? What about the blood-brain barrier?

Response: When we refer to the matrisome database, we are referring to the bioinformatics-based human database reported in Naba et al 2012. We have amended the text to make it clear we have used the human database and provided the reference on page 5 and 19.

We have added the number of differentially expressed proteins in MDA-bone versus MDA-lung to the bottom of the figure panel, which is now **Figure 1E**.

We have added a sentence referring to the blood-brain barrier on page 6, which may block secreted factors from the primary tumour from 'reprogramming' metastasised brain cells or those in the brain microenvironment.

3. Page 6, results, what makes PEG hydrogels biomimetic? Explain this here in more detail again (or establish the term 'biomimetic' in the introduction) and why this is important for MSCs. Why was perlecan, a large proteoglycan found in the vascular ECM, chosen as representative ECM factor in figure 2A? Is the medium for the secretome cultures different to the actual MSC culture medium? Was the secretome medium replaced during the 7-day treatment period? Why was a 7-day treatment chosen? Transcriptomic changes will occur much earlier. The secretome analysis was performed after 24 hours using serum-free conditions, while the effect of the conditioned medium on MSCs was measured after 7 days.

Response: We have now described our PEG hydrogel in more detail on page 7 and explained what makes it biomimetic. The biomimetic character of the hydrogels is important not only for MSCs but for many other cell types, because PEG is an inert material and requires biomimetic modifications to enable fundamental cell processes (such as cell adhesion and migration).

Perlecan (HSPG-2) was chosen because it is one of the ECM proteins that is deposited by MSCs into hydrogels at very early time points (already present in the extracellular space after 24h). In fact, MSCs re-shape PEG hydrogels strongly by ECM deposition and other key ECM proteins produced by MSCs include Fibronectin and Collagen type 1 (see Blache et al., EMBO Reports, 2018). Given the close relationship (overlap) between MSCs and pericytes it is not surprising that MSCs produce a lot of the vascular basement membrane ECM protein perlecan.

Breast cancer secretomes (conditioned media CM) were collected in FBS-free DMEM/F12 and 20x concentrated. MSCs were cultured in MEM-a (2% FCS) supplemented with 5% of 20x CM to reach a 1x breast cancer secretome in MSC medium. We renewed 50% of the prepared medium daily. This information is given in the materials and methods section on page 20, which for the initial submission was a supplementary file but now is present in the main manuscript.

The secretome analysis was performed after 24h because this is time point when we collected the secretome from the breast cancer cells and added it to the MSC cultures. Therefore, we know the composition of the secretomes that we added to the MSCs. We measured the effect of the breast cancer secretome on MSCs after 7 days to cover a wide range of TA-MSCs, which potentially also includes an autocrine response loop upon education by breast cancer secretomes.

4. Page 7, results, a PTK inhibitor or specific blocking antibody could be used to validate the changes in the kinase profile in MSCs. Include references for statement 'are known to be involved in cancer and angiogenesis'. The advantage of RNAseq over MS/protein analysis should be better explained as this technique also helps to identify novel transcripts and genes that might not encode for functional proteins.

Response: Experiments with PTK inhibitors or blocking antibodies would certainly be interesting experiments. However, they would go beyond the scope of this manuscript, which was to provide a global picture of TA-MSC activation by cancer secretomes. Nevertheless, we believe that our manuscript is a valuable research resource for future studies focusing on RTK/oncogene changes in non-tumor/stromal cells in the tumor microenvironment; e.g. in CAFs or TA-MSCs. As mentioned above, we have added in an additional phosphopeptide analysis of our data, and validated our mass spectrometry and RNA sequencing results.

We have included references for ephrin receptors involved in cancer on page 7, and we have expanded the sentence about RNAseq on page 8.

5. Pages 9-10, results, include references for statement 'aerobic glycolytic activity is a hallmark of cancer and is known as the Warburg effect'. Why is COMP an important factor in bone or lung metastasis?

Response: We have included the required references for the Warburg effect on page 10.

To our knowledge, it is not known why COMP is an important factor in bone or lung metastasis, which is why we found it intriguing that it was highlighted in our analysis. We have added 2 additional references (page 11) from studies that have identified upregulated stromal COMP in ovarian cancer metastasis and have shown that COMP promotes lung metastasis in breast cancer, which are in support of our data.

6. The conclusions need to be revised and should include some therapeutics that are important for targeting TME components that promote breast cancer and metastasis. What is the stiffness of MSC-seeded hydrogels? Does it change with treatment using the different secretomes? PEG hydrogels are an established model for 3D cultures and disease platforms as well as multi-omics and multi-level analyses. Different PEG matrices are used by various different researchers around the world and more cancer-specific references should be included for people readers which are not familiar with these hydrogels.

Response: The stiffness of our used hydrogels is 470 Pa (Storage modulus as assessed by *in situ* rheology; Blache et al. EMBO Reports, 2018). We have added this information to the manuscript on pages 4 and 20.

It has been described that the stiffness of PEG hydrogels slightly changes in pericellular area of MSCs (Ferreira et al., Nature Communications 2018). It is possible that in response to different secretomes MSCs modify their environment differently and by this change the pericellular stiffness to a different degree. The assessment of local changes in stiffness could principally be conducted by atomic force microscopy. However, we expect such evaluations to be extremely difficult since changes would likely be much smaller than in the study by Ferreira.

We have added further cancer-specific references applying PEG hydrogels as 3D culture platforms on page 3.

In the introduction on page 3, we have stated that 'several agents targeting tumour stroma are in clinical trials' and provided a reference.

Minor comments:

1. In the abstract, 'bioengineered synthetic' should rather read 'synthetic'; use at the end of this sentence 'bioengineered 3D microenvironment'; delete 'PEG'. What specific anti-cancer therapeutics target cancer-supporting cells? What about anti-metastatic treatments? The last sentence should be extended.

Response: We have made text changes to the abstract as suggested. We have changed 'cancer support cells' to 'tumour stroma'. We do not have space in the abstract to mention specific

therapies, however, we have added a new sentence to the end of the first paragraph in the introduction.

2. Presumably, the MDA-bone/lung/brain cells are human. How were they derived? The authors should also refer to the original paper in which these cell lines have been established.

Response: Yes, the MDA-MB-231 sublines are human. They have been generated by repeated, organ-tropic *in vivo* selection of MDA-MB-231 cells that had been injected to the left ventricle of nude mice. We have added this information to the manuscript on page 5 together with the original references.

3. Page 3, introduction, use the abbreviation 'TME' for tumour microenvironment. What are 'confounding signals'? Use 'cytocompatible' instead of 'cell-friendly'. Explain abbreviations upon first usage, for example MMP.

Response: We have followed these suggestions and changed the manuscript text accordingly. Confounding signals can be proteins (growth factors, matrix proteins) present in ill-defined ECM-derived hydrogels materials such as in matrigel. For instances, it is not surprisingly but still impressive that even the growth-factor reduced version of matrigel consists of more than 1000 different proteins (Hughes et al. Proteomics, 2010), which could have a huge influence on molecular signalling data.

4. Pages 7-9, results, reword 'milder' and 'mild'.

Response: We have removed these words from the manuscript.

5. Legends for figures 1 and 2 need to be shortened.

Response: We have shortened the figure legends.

6. Page numbers in the supplementary methods are missing.

Response: The supplementary methods now have been integrated into the main manuscript and include page numbers.

May 21, 2019

RE: Life Science Alliance Manuscript #LSA-2019-00304-TR

Dr. Martin Ehrbar
University of Zurich
Obstetrics
Schmelzbergstrasse 12
Zurich, Zurich 8091
Switzerland

Dear Dr. Ehrbar,

Thank you for submitting your revised manuscript entitled "Mesenchymal stromal cell activation by breast cancer secretomes in synthetic 3D hydrogels". As you will see, the reviewers appreciate the introduced changes and we would be happy to publish your paper in Life Science Alliance pending final revisions necessary to substantiate the validation in Fig 1F as suggested by reviewer #2. Please also address the remaining comments of this reviewer when preparing the final manuscript files.

A. FINAL FILES:

B. MANUSCRIPT ORGANIZATION AND FORMATTING:

Sincerely,

Reviewer #1 (Comments to the Authors (Required)):

The authors have revised the manuscript based on suggestions offered by the reviewers. The detailed responses and additional data included in the revised manuscript are appropriate and the revised manuscript will be of interest to readers of LSA.

It is acceptable for publication

Reviewer #2 (Comments to the Authors (Required)):

The authors have submitted a revised version of the manuscript, which has addressed most of the previous concerns. I have appreciated the effort that the authors have made to provide some links between the different analyses and to validate some of the hits of the secretome and RNAseq analyses. The new data have improved the robustness of the story. However, I would ask the authors to include a couple of information in Figure 1F:

1. One example of protein upregulated in MCF7 cells compared to the MDA-WT line, for example COL1A1 for which there are good antibodies.
2. Add the quantification of the western blot in Figure 1F, possibly using replicate experiments.

Minor observations:

Page 6: "Similarly, the western blot analysis also confirmed the differences between MDA sub-lines". It is true that WB confirmed the regulation of some proteins, but not all (SERPINE1 and FN1).

Please, tone down this sentence.

Page 8: the new sentence highlighted in yellow is not very clear.

Page 12: "growth factors and cytokines are notoriously difficult to detect by label-free methods...". Why do the authors think that GFs and cytokines are particularly difficult to detect with label-free methods? I think that it's not the protein quantification strategy that was not suitable to detect those factors, but rather the fact that they started with low amount of material and that the MS instrument that they have used, a QExactive (which is quite old), was not sensitive enough. I would remove the Label-free part and add the use of new generation MS instruments to the options to improve their data.

Figure 1F legend: what does stain-free protein gel mean? Is it a coomassie or red ponceau staining?

Figure 1E: I would highlight with an asterisk the proteins validated in panel F.

Reviewer #3 (Comments to the Authors (Required)):

The authors have addressed most of the points raised by the reviewers and carried out some validation experiments, including RT-qPCR, Western blot and migration analyses, more or less convincing. An additional network analysis has now been included into the main manuscript. They have improved their referencing and removed some of their rather speculative conclusions.

May 22, 2019

RE: Life Science Alliance Manuscript #LSA-2019-00304-TRR

Dr. Martin Ehrbar
University of Zurich
Obstetrics
Schmelzbergstrasse 12
Zurich, Zurich 8091
Switzerland

Dear Dr. Ehrbar,

Thank you for submitting your Research Article entitled "Mesenchymal stromal cell activation by breast cancer secretomes in synthetic 3D hydrogels". Thank you for having attempted to show COL1A1 upregulation in MCF7 vs MDA-WT secretomes. I understand that you failed to detect any protein, indicating expression below the detection limit. I appreciate the other introduced changes, and it is a pleasure to let you know that your manuscript is now accepted for publication in Life Science Alliance. Congratulations on this interesting work.

*****IMPORTANT:** If you will be unreachable at any time, please provide us with the email address of an alternate author. Failure to respond to routine queries may lead to unavoidable delays in publication.*******

DISTRIBUTION OF MATERIALS:

Again, congratulations on a very nice paper. I hope you found the review process to be constructive

and are pleased with how the manuscript was handled editorially. We look forward to future exciting submissions from your lab.

Sincerely,
